# ZnONPs Alleviates Salt Stress in Maize Seedlings by Improving Antioxidant Defense and Photosynthesis Potential

**DOI:** 10.3390/plants14193104

**Published:** 2025-10-09

**Authors:** Siqi Sun, Xiaoqiang Zhao, Xin Li, Meiyue He, Jing Wang, Xinxin Xiang, Yining Niu

**Affiliations:** State Key Laboratory of Aridland Crop Science, College of Agronomy, Gansu Agricultural University, Lanzhou 730070, China; sunsiqi1215@163.com (S.S.); m18214360633@163.com (X.L.); hemeiyue0513@163.com (M.H.); 15138079530@163.com (J.W.); 18403693151@163.com (X.X.); niuyn@gsau.edu.cn (Y.N.)

**Keywords:** NaCl stress, ZnONPs, antioxidant defense, photosynthesis parameters, comprehensive evaluation, cluster analysis

## Abstract

Salt stress is a significant environmental factor that inhibits maize growth and development, severely affecting yield formation. Interestingly, nanomaterials, particularly ZnONPs, can enhance resistance to various stresses and support healthy crop growth. However, the effects of ZnONPs on maize under salt stress remain unclear. This study investigates the effect of foliar and seed exposure to zinc oxide nanoparticles (ZnONPs) on reducing NaCl-induced salt stress in two maize inbred lines (NKY298-1 and NKY211). Over a period of seven days, under 120 mM NaCl, we measured growth, reactive oxygen species (ROS), malondialdehyde (MDA), membrane stability index (MSI), water status (relative water content, RWC), photosynthetic pigments and parameters, selected photosynthetic enzymes, and antioxidant enzyme activities. Then, we propose four composite indices, including stress improvement index (SII), alleviation capacity index (ACI), comprehensive improvement effects (CIE), and comprehensive alleviation capacity (CAC), to rank the effectiveness of ZnONP doses. The findings suggested that 50–100 μM ZnONPs significantly mitigate salt damage, with optimal doses varying by genotype (50 μM for NKY211 and 100 μM for NKY298-1). Notably, the study’s originality lies in its side-by-side composite scoring across 26 traits in two maize genotypes’ seedlings. In conclusion, the findings will provide a new idea for research on the molecular mechanism by which exogenous ZnONPs application improves the salt tolerance of maize seedlings.

## 1. Introduction

It is estimated that over 800 million hectares of arable land worldwide are affected by salinity (typically caused by NaCl accumulation), accounting for over 6% of total land area [1]; saline–alkali soils account for 25% of arable land in China and remain underutilized [2]. With increasing salinization, 50% of cultivable lands will be salinized by 2050 [1,3], resulting in a reduction in arable lands, which will inevitably influence species distribution and sustainable agricultural production. Thus, salinity has become one of the most severe environmental stresses that inhibits plant growth and development, and a 500 mM NaCl solution causes an approximate 65% yield loss in wheat (*Triticum aestivum* L.) [4].

Maize (*Zea mays* L.) is a globally important cereal crop that is widely used for food, feed, and bioenergy, with an annual production of 1200 million tons [5]. The early growth stage of maize is extremely sensitive to salinity; its plants are affected by salt levels of more than 250 mM NaCl, which can restrict growth and produce severe withering [6]. High concentrations of Na^+^ and Cl^−^ in the soil lead to osmotic stress, which decreases root water absorption and inhibits the cell expansion of maize [7]; meanwhile, in maize grown under NaCl stress, there is disturbed ionic balance, disrupting the Na^+^/K^+^ ratio and influencing nutrient transport [8,9]. Moreover, ionic toxicity occurs in NaCl-stressed maize, which causes oxidative damage and excessive generation of reactive oxygen species (ROS) [10]; these damages then bring about various enzyme functionality disruptions, inhibition of photosynthesis, and, consequently, premature senescence [11]. In addition, previous studies have confirmed that maize possesses significant intraspecific genetic variations in salt tolerance [12,13].

As is well known, nanomaterials such as nanoparticles (NPs) can facilitate a targeted delivery of nutrients and genetic materials to plants, which is difficult for traditional materials due to biological barriers in plants [14], along with augmenting plants’ tolerances and maintaining their health growth. For instance, applying 200 mg kg^−1^ silica (Si)NPs significantly boosted the growth and productivity of cucumber (*Cucumis sativus* L.) plants in a water deficit and under salinity stresses by balancing their nutrient uptake [15]. A 150 μM bio-selenium (Se)NPs treatment significantly alleviated 150 mM NaCl stress in rapeseed (*Brassica napus* L.) by enhancing the seeds’ water absorption capacity and antioxidant system during the early seedling stage [16]. We used 10 mg/L zinc oxide NPs (ZnONPs) to stimulate the growth of sorghum (*Sorghum bicolor* L.) under salt stress and also to improve the distribution of elements [17]. The negative effects of NaCl on lupin (*Lupinus termis*) were alleviated by 60 mg/L ZnONPs, as it enhanced photosynthetic pigments, regulated osmotic regulation, and reduced the contents of malondialdehyde (MDA) and Na [18]. Compared with conventional ZnSO_4_ fertilizer, nano zinc oxide has distinctive physicochemical characteristics and increases the availability of micronutrients for plants. However, ZnONPs has not yet been applied in maize cultivation, which is an important strategic crop that is sensitive to salt.

Therefore, we conducted experiments by applying five concentrations of ZnONPs (Nanoparticle, ultrafine unit with dimensions measured in nanometers (nm; 1 nm = 10^−9^ m)) suspensions to two maize genotypes seedlings (NKY298-1 and NKY211) at a concentration of 120 mM NaCl. It is expected that an appropriate concentration can be obtained to alleviate the damage caused by salt stress to maize.

## 2. Results

### 2.1. Combined Analysis of Different Treatments and Genotypes of Various Traits in Maize

Factor analysis of variance (ANOVA) proved that the tested variables (maize genotype, NaCl, and ZnONPs) and their physiological and molecular interactions with 26 measured traits were significant in two maize genotype seedlings (Table 1). These data demonstrated that the resistance of maize to salt stress was controlled by factors such as its own genetic structure, environment, and interaction. Our results also indicated that the interaction between the genotype and ZnONPs had a greater impact on these traits.

### 2.2. Phenotypic Changes in Two Maize Genotypes Under Different Treatments

The growth and development of two maize seedlings were generally significantly inhibited when treated with salt stress (120 mmol/L NaCl) for 7 days (Figure 1). Compared to CK treatment, NaCl stress had a significantly negative impact on the phenotypic growth condition of two maize seedlings, especially the root fresh weight (RFW), which dropped by 68.27% and 34.32% in NKY298-1 and NKY211, respectively (Figure 1D). Moreover, under different ZnONPs’ (ZnO nanoparticles) concentrations application treatments, the aboveground and underground parts of the two maize seedlings exhibited different reactions. For NKY298-1, the seedling fresh weight (SFW) was more sensitive to the addition of 100 μmol/L ZnONPs, which increased it by 68.62% compared to NaCl stress (Figure 1C), and for NKY211, the seedling length (SL) and SFW were more sensitive to the concentration of 50 μmol/L, which increased them by 17.19% and 21.21%, respectively, compared to NaCl stress (Figure 1A,C). Then, for the underground parts, NKY298-1’s RL and RFW were more sensitive to the concentration of 100 μmol/L, which increased them by 8.59% and 289.17%, respectively, compared to NaCl stress (Figure 1B,D). The NKY211 genotype showed a significantly positive change under 50 μmol/L ZnONPs supplement with an increase of 27.85% in RL and 30.05% in RFW (Figure 1B,D). The better alleviation effects on NaCl stress in the aboveground and underground parts of the NKY298-1 and NKY211 genotypes were the application of 100 μmol/L and 50 μmol/L ZnONPs suspensions, respectively.

### 2.3. The Responses in Oxidative Stress and Membrane Stablity of Two Maize Genotypes Under Different Treatments

A previous study showed that oxidative stress can affect the salt stress tolerance of plants [19]. In the current study, under NaCl stress, both the NKY298-1 and NKY211 genotypes were in a significant state of oxidative stress: namely, the O_2_^•−^ content increased by 66.71% and 51.92%, respectively, (Figure 2A), the H_2_O_2_ content increased by 74.79% and 59.74%, respectively, (Figure 2B), and the malondialdehyde (MDA) content increased by 26.28% and 23.33%, respectively, in two maize seedlings (Figure 2C). Zhao et al.’s research [20] showed that the MDA content and MSI of plants can reflect the degree of lipid peroxidation and membrane stability under different environmental stresses. In the current study, the membrane stability index (MSI) and relative water content (RWC) decreased by 30.29% and 26.47% in MSI, respectively, and 31.34% and 27.88% in RWC, respectively, under NaCl stress in two maize seedlings (Figure 2D,E). For the NKY298-1 genotype, 100 μmol/L ZnONPs application significantly decreased the O_2_^•−^ content (33.41%), H_2_O_2_ content (32.42%), and MDA content (45.88%) (Figure 2A–C). Meanwhile, there was an increase in the MSI (31.59%) and RWC (27.77%) (Figure 2D,E). However, for the NKY211 genotype, the application of 50 μmol/L ZnONPs had a more significant function, with a 14.55% decrease in O_2_^•−^ content, a 27.21% decrease in H_2_O_2_ content, a 27.21% decrease in MDA content, and an 18.29% increase in RWC (Figure 2A–C,E), but the MSI changed more significantly at a concentration of 100 μmol/L with 11.86% (Figure 2D). These data changes indicated that the oxidative stress induced by NaCl stress can damage the membrane stability of maize. For different genotypes’ maize seedlings, the addition of appropriate concentrations of exogenous ZnONPs can alleviate the damage caused by NaCl stress.

### 2.4. The Changes in Photosynthetically Related Parameters of Two Maize Genotypes Under Different Treatments

Chlorophyll is an indispensable cofactor for photosynthesis; maintaining a normal chlorophyll state and level is crucial for photosynthetic efficiency and carbon sequestration, directly affecting the growth and development of plants under various abiotic stresses [21]. Salt stress can also affect the activities of various key photosynthetic enzymes in maize, such as adenosine triphosphate synthase (ATPase, play a key role in the light-dependent stage of photosynthesis), phosphoenolpyruvate carboxylase (PEPCK, mainly functions to fix carbon dioxide into four-carbon compounds and transport the fixed carbon dioxide to the vascular bundle sheath cells through the “C-pump” mechanism, promoting the efficiency of photosynthesis), and ribulose-1, 5-diphosphate carboxylase (RuBPase, a key enzyme in the dark reaction of photosynthesis, and its main function is to catalyze the combination of CO_2_ and RuBP (ribulose-1, 5-diphosphate) to form 3-phosphoglyceric acid (3-PGA), which is the initial step of the Calvin cycle), thereby inhibiting CO_2_ fixation and photosynthetic efficiency [22,23,24]. In the current study, NaCl stress led to a significant reduction in the contents of Ca (chlorophyll a, with 23.18% and 12.41%, respectively), Cb (chlorophyll b, 14.48% and 9.15%, respectively), Cab (chlorophyll a+b, 20.10% and 11.25%, respectively), Car (carotenoid, 22.55% and 20.97%, respectively), and the ratio of Ca/Cb (chlorophyll a:b ratio, 10.37% and 3.70%, respectively) in NKY298-1 and NKY211 (Figure 3A–E). But the ratio of Cab/Car (chlorophyll a+b content: carotenoid content ratio) increased by 3.17% in the NKY298-1 genotype and 12.77% in the NKY211 genotype under NaCl stress, compared to CK treatment (Figure 3F). At the same time, NaCl stress also led to a decrease in the content of photosynthetic enzymes in the two maize genotypes, especially the PEPCK (with the increase of 54.29% and 50.98% in NKY298-1 and NKY211, respectively) (Figure 3H). This indicated that NaCl stress significantly affected the photosynthetic capacity of maize, and different genotypes were affected differently. For the NKY298-1 genotype, 50 and 100 μmol/L ZnONPs both had a more significant impact on the content of photosynthetic pigments and photosynthetic enzymes, but for the NKY211 genotype, the influence of ZnONPs with a concentration of 50 μmol/L was more significant (Figure 3A–I).

A high salt concentration in soil can lead to osmotic stress, causing physiological problems such as cell expansion and decreased stomatal conductance. It also had a negative impact on photosynthesis, biomass accumulation, and yield, and impaired the absorption of essential nutrients [25,26,27]. In the current study, NaCl stress negatively impacted the photosynthetic parameters, such as Pn (net photosynthetic rate), Gs (stomatal conductivity), Ci (intercellular CO_2_ concentration), and Tr (transpiration rate) (Figure 4A–D). Compared to CK, NaCl stress caused the decreases in the Pn (38.25% and 18.14% in NKY298-1 and NKY211, respectively), Gs (37.22% and 58.03% in NKY298-1 and NKY211, respectively), and Tr (49.75% and 49.97% in NKY298-1 and NKY211, respectively) (Figure 4A,B,D). Meanwhile, it increased the Ci (44.43% and 55.94% in NKY298-1 and NKY211, respectively) (Figure 4C). Overall, the photosynthetic parameters of NKY211 were less affected by NaCl stress than those of NKY298-1. In addition, when it came to alleviating the damage caused by NaCl stress to photosynthetic indicators, the NKY298-1 had a better effect when treated with 100 μmol/L ZnONPs, while the NKY211 had a better effect when treated with 50 μmol/L ZnONPs.

### 2.5. The Changes in Antioxidant Enzyme Activity of Two Maize Genotypes Under Different Treatments

The overexpression of miR528 could widely improve the salt tolerance of rice by downregulating the *AO* gene encoding L-ascorbic acid oxidase [28,29]. Different salt concentrations had a significant impact on the activity of antioxidant enzymes; under salt stress, antioxidant enzymes enhance the protective effect of cells against oxidative stress [30]. In the current study, NaCl stress caused varying degrees of damage to two maize genotypes seedlings; the increase in antioxidant enzyme activity in NKY298-1 under NaCl stress was higher than that in NKY211 (Figure 5A–D), which may indicate that NKY298-1 had a higher tolerance to salt stress. For the NKY298-1 genotype, the increase in antioxidant enzyme activity was 3.12% (SOD, superoxide dismutase activity); 1.45% (POD, peroxidase activity); 1.30% (CAT, catalase activity); and 2.35% (APX, ascorbate peroxidase activity), respectively, and the antioxidant enzyme activity in the NKY211 genotype increased by 1.46% (SOD); 1.41% (POD); 1.36% (CAT); and 2,76% (APX), respectively, under NaCl stress (Figure 5A–D). Then, the ZnONPs application enhanced the activity of the four enzymes by 1.35% (SOD), 1.06% (POD), 1.40% (CAT), and 1.31% (APX), respectively, in two maize genotypes seedlings under NaCl stress treatment. For antioxidant enzymes, the application of 100 μmol/L ZnONPs was more effective to alleviate the damage caused by NaCl stress for both the NKY298-1 and NKY211 genotypes.

### 2.6. Cluster Analysis

To evaluate the alleviation effect of different concentrations of ZnONPs on NaCl stress in maize, three cluster-analysis methods were adopted to qualitatively identify the alleviation effects of two maize inbred lines under all treatments. All three methods obviously classified the two maize genotypes under different treatments into four groups (Figure 6A–C). NKY298-1 and NKY211 under CK treatment were the I group, indicating the growth status and physiological indicators of the two maize inbred lines were the best under this treatment; then, the NKY298-1 (S + ZnONPs-100), NKY298-1 (S + ZnONPs-50), and NKY211 (S + ZnONPs-50) were the II group, which indicated that the growth conditions of the two maize seedlings under the stimulation of the two concentrations of ZnONPs were better than other concentrations. Moreover, according to the results of both the hierarchical clustering heatmap and the circular clustering heat map, NKY298-1 (S), NKY211 (S), NKY211 (S + ZnONPs-150), and NKY211 (S + ZnONPs-180) were consistently divided into group IV, suggesting that the growth status of the two maize genotypes was the worst under NaCl stress and the two concentrations of ZnONPs, resulting in the alleviation effect of the two concentrations on the damage caused by NKY211 under NaCl stress being very weak. Overall, for NKY298-1, the concentration of 100 μmol/L and 50 μmol/L had the best alleviation effect, and for NKY211, 50 μmol/L ZnONPs application had the best alleviation effect.

### 2.7. Comprehensive Evaluation

To comprehensively evaluate the ability of different concentrations of ZnONPs to alleviate salt stress in two maize genotypes seedlings, we introduced four definitions: namely, stress improvement index (SII), alleviation capacity index (ACI), comprehensive improvement effects (CIE), and comprehensive alleviation capacity (CAC). The range of SII of NKY298-1 was smaller than that of NKY211 (Figure 7A), and it was the same in ACI (Figure 7B). Among the 26 traits, exogenous ZnONPs application had the best effect on improving the stress of PEPCK in the NKY298-1 and NKY211 genotypes (Figure 7A), while Cab/Car in NKY298-1 and Ca/Cb in NKY211 had the best ACI effect (Figure 7B). In addition, the 100 μmol/L and 50 μmol/L ZnONPs solutions had the best CIE in NKY298-1 and NKY211, respectively; (Figure 7C) similarly, the two maize genotypes exhibited the best CAC at the same two concentrations (Figure 7D).

## 3. Discussion

Salt stress is the main environmental factor in soil that inhibits and negatively affects plant growth, development, and yield formation [31]. Luckily, there were already some strategies to alleviate soil salt stress in some plants. For example, the inoculation of rhizobia had the effect of alleviating the salt stress damage of alfalfa (*Medicago sativa* L.) [32]; the synergistic effect of H_2_S and NO alleviated the negative impact of salinity by regulating the antioxidant defense of bitter gourd (*Momordica charantia* L.) seedlings and reducing oxidative damage [33]; nano-selenium and zinc oxide had an alleviation effect on salt stress in rice (*Oryza sativa* L.) [34]; and the application of Si (silicon) reduced the effects of NaCl on abscisic acid (ABA) and jasmonate (JA), and decreased the effects of NaCl-mediated salt stress on maize plants [35]. However, at present, there are very few studies on using nanomaterials to alleviate maize salt stress. Therefore, in the current study, we selected five ZnONPs concentrations to treat two maize genotypes seedlings under NaCl stress, and observed changes in the antioxidant enzymes (i.e., SOD, POD, CAT, and APX) activity, the content of substances related to oxidative stress (such as O_2_^•−^, MDA, and H_2_O_2_), and the changes in photosynthetically related substances (such as Pn, Gs, Ci, Tr, PEPCK, PPDK, Ca, Cb, and so on). Meanwhile, we also comprehensively evaluated the alleviation effects of maize seedlings under salt stress by using multiple clustering methods and membership function methods at different concentrations of ZnONPs.

The research of Rozita et al. indicated that salt stress induces oxidative damage to garlic (*Allium sativum* L.) plants [36]. Just like this, we found that the content of O_2_^•−^, H_2_O_2_, and MDA increased by 59.32%, 67.27%, and 24.81%, respectively, under NaCl stress (Figure 2A–C), and the MSI and RWC decreased by 28.38% and 29.61%, respectively, under NaCl stress in two maize seedlings (Figure 2D,E). We found that the antioxidant enzyme activity increased, especially the SOD (2.29%), compared to CK (Figure 5A–D). At the same time, the growth of aboveground and underground parts all decreased, especially the RFW (51.30%) (Figure 1A–D); this is consistent with the research results for arabidopsis (*Arabidopsis thaliana* (L.) Heynh.) by Lin et al. [37]. These changes indicated that salt stress caused oxidative stress responses in plants, thereby affecting their growth.

As is well known, nanoparticles (NPs) can promote the targeted delivery of nutrients and genetic material into plants and enhance the tolerance of plants [14]; meanwhile, studies have shown that nanoparticles (NPs) are an effective method to alleviate salt stress [38]. According to previous studies, foliar spraying of 100 mg/L ZnONPs alleviated high salt stress in cotton (*Gossypium hirsutum* L.) by adjusting the Na+/K+ ratio and regulating antioxidant capacity [38]; the application of ZnONPs (0.12 g POT^−1^) significantly increased the contents of chlorophyll A and B in wheat (*Triticum aestivum* L.), and it could be used as a substitute for conventional zinc fertilizer in salt-damaged soil to achieve better crop yields [39]; foliar spraying of 75 and 150 mg/L ZnONPs alleviated the effect of 150 mM NaCl salt stress on the growth of tomato plants (*Solanum lycopersicum* L.) [40]; and foliar spraying of 100 mg/L ZnONPs had a beneficial effect on promoting the growth of radish plants (*Raphanus sativus* L.), and reduced the adverse effects of NaCl stress [41]. Just like the results of Qian et al. [38], in our experimental results, it was found that ZnONPs reduced the average levels of O_2_^•−^, H_2_O_2_, and MDA with 14.44%, 21.07%, and 25.94% in two maize seedlings (Figure 2A–C) and by increasing the activity of major antioxidant enzymes (such as SOD, POD, CAT, and APX with 1.35%, 1.06%, 1.40%, and 1.31%, respectively) in two maize seedlings (Figure 5A–D), to significantly improve salt-induced oxidative stress. Meanwhile, the MSI and RWC had significantly increased in two maize seedlings by 17.26% and 14.32% under ZnONPs treatment. Different concentrations of ZnONPs had different alleviation effects on two maize genotypes; for the NKY298-1 genotype, 100 μmol/L ZnONPs solutions had a more significant alleviating effect on most traits, and for the NKY211 genotype, 50 μmol/L and 100 μmol/L ZnONPs solution had a more significant alleviating effect on most traits.

In addition to improving antioxidant enzyme defense, ZnONPs also played a significant role in enhancing photosynthetically related traits. Research by Wang et al. showed that high salt content inhibited photosynthetic carbon assimilation and ZnONPs promoted the tolerance of photosynthetic devices to subsequent salt stress, enhancing the efficiency of photosynthetic electron transfer and sucrose biosynthesis in leaves under salt stress [42]. In the current study, 100 μmol/L and 50 μmol/L ZnONPs application could significantly increase most photosynthetic traits in the NKY298-1 and NKY211 genotypes, respectively, especially the photosynthetic enzymes such as NADP-ME, PEPCK, and PPDK (Figure 3G–I), and the photosynthetic pigments had also been improved (Figure 3A–F).

Then we found that NKY298-1 (S + ZnONPs-100), NKY298-1 (S+ ZnONPs-50), and NKY211 (S+ ZnONPs-50) were clustered in group II, using three different clustering methods, which was also the group with the best alleviation effects (Figure 6A–C). In addition, the improvement effect of ZnONPs application on different traits of two maize genotypes was also different (Figure 7A,B). Finally, we used the membership function method to comprehensively evaluate the alleviation ability and improvement effect of different ZnONPs concentrations’ application on 26 traits of two maize genotypes under NaCl stress. Then, we found that 100 μmol/L ZnONPs application had the best comprehensive improvement effect (CIE) and comprehensive alleviation capacity (CAC) for NKY298-1, and 50 μmol/L ZnONPs application for NKY211 (Figure 7C,D).

In conclusion, 50–100 μmol/L ZnONPs application (50 μM for NKY211 and 100 μM for NKY298-1) had the best alleviation effect on different maize genotypes under NaCl stress. However, at present, due to issues such as price and variety adaptability, the application of field maize cultivation has been affected. Therefore, the promotion of ZnONPs application is of vital importance in corn production.

## 4. Materials and Methods

### 4.1. Maize Materials and Experimental Design

The two elite genotypes maize inbred lines NKY298-1 and NKY211 from the Longxi experimental station in Gansu, China (34.97° N, 104.40° E, 2074 m altitude) were used in this study. Uniform seeds were first sterilized 0.5% (*v*/*v*) sodium hypochlorite solution for 10 min and then rinsed five times with double-distilled water (ddH_2_O). The sterilized seeds were soaked in darkness for 24 h in ddH_2_O, NaCl salt solution, and five concentrations of exogenous ZnONPs (Yuanye Biotechnology Co., Ltd., Shanghai, China; CAS: 1314-13-2 V33683-100g ≥ 99%, powder, ≤100 nm) solution (i.e., 10, 50, 100, 150, and 180 μmol/L), respectively. Each of the above ZnONPs solutions was mixed separately with sterilized vermiculite uniformly, in a proportion of 100 mL: 500 g to prepare the seed sowing matrix. The seeds of each inbred line were soaked in the corresponding concentrations of ddH_2_O, NaCl salt solution and five concentrations of ZnONPs solutions for 24 h, respectively, then evenly sowed on the sowing substrate in plastic boxes (13 cm diameter × 11 cm high) with 10 seeds in each pot, and covered with the corresponding sowing substrate, maintaining a sowing depth of 3 cm. Each treatment was repeated three times, and all the potted plants were placed in an intelligent artificial climate room, in which the culture conditions were set as 12/12 h light/dark cycle, 20 ± 0.5 °C constant temperature, 300 µMm^−2^ S^−1^ light intensity, and 60% relative humidity for seven days. We sprayed 20 mL of the corresponding solution on the leaves of each can every two days.

### 4.2. Phenotypic and Physiological Assays

After 7 days of seed germination under the seven treatments in Table 2. (i.e., CK: 0 mmol/L NaCl solution + 0 μmol/L ZnONPs in 20 °C environment; S: 120 mmol/L NaCl solution in 20 °C environment; S + ZnONPs-10: 120 mmol/L NaCl solution + 10 μmol/L ZnONPs in 20 °C environment; S + ZnONPs-50: 120 mmol/L NaCl solution + 50 μmol/L ZnONPs in 20 °C environment; S + ZnONPs-100: 120 mmol/L NaCl solution + 100 μmol/L ZnONPs in 20 °C environment; S + ZnONPs-150: 120 mmol/L NaCl solution + 150 μmol/L ZnONPs in 20 °C environment; S + ZnONPs-180: 120 mmol/L NaCl solution + 180 μmol/L ZnONPs in 20 °C environment.), the seedlings were quickly washed off with ddH_2_O, and three seedlings with the same growth were selected to measure growth parameters from each pot under different treatments, including the seedling length (SL), root length (RL), seedling fresh weight (SFW), and root fresh weight (RFW). The separated seedlings and roots under each treatment were frozen in liquid nitrogen immediately and stored at −80 °C for physiological analysis.

And some physiological indicators were determined using the Solarbio biochemical kit, including O_2_^•−^ content (Cat: BC1290), H_2_O_2_ content (Cat: BC3595), MDA content (Cat: BC6415), NADP-ME content (Cat: BC1125), PEPCK content (Cat: BC3315), PPDK content (Cat: BC5855), SOD activity (Cat: BC5165), POD activity (Cat: BC0095), CAT activity (Cat: BC0205), and APX activity (Cat: BC0225), and using a multi-function microplate reader (Syner622 gyHTX; BioTek Instruments, Inc., Winooski, VT, USA), following the manufacturer’s instructions, respectively.

There were four photosynthetic parameters, including the net photosynthetic rate (Pn; µmol CO_2_·m^−2^·s^−1^), the intercellular CO_2_ concentration (Ci; µmol CO_2_·m^−2^·s^−1^), the stomatal conductance (Gs; µmol CO_2_·m^−2^·s^−1^), and the transpiration rate (Tr; µmol CO_2_·m^−2^·s^−1^). Using the LI-6400XT (LI-COR; Portable photosynthesis measurement system, Lincoln, NE, USA, Bioscience Co., LTD, Seongnam-si, Republic of Korea), the illuminance in the leaf cavity was set at 1000 µM/(m^2^·s), the CO_2_ concentration at 400 µM·M^−1^, and the temperature at 25 °C.

For the chlorophyll content, we referred to the method of Qi et al. [32]. For the determination, 0.1 g fresh leaves of NKY298-1 and NKY211 seedlings were taken, respectively, for each treatment, and soaked in 10 mL 95% alcohol for 48 h. Then, they were centrifuged at 12,000 rpm (centrifuge 5425/5425 R; Eppendorf, Hamburg, Germany, 10 min), utilizing the multifunctional enzyme-labeled model (SynergyHTX; BioTek Instruments, Inc., Winooski, VT, USA), with wavelengths of 665 nm, 649 nm and 470 nm. The concentration calculation formulas for chlorophyll a (Ca), chlorophyll b (Cb), and carotenoids (Car) are as follows:Ca = 13.95 × A665 − 6.88 × A649;(1)Cb = 24.96 × A649 − 7.32 × A665;(2)Car = 1000 × A470 − 2.05 × Ca − 114.8 × Cb.(3)

The calculation formula for pigments content is the following:Chl a content (mg g^−1^) = Ca × Vt × n/FW × 1000;(4)Chl b content (mg g^−1^) = Cb × Vt × n/FW × 1000;(5)Car content (mg g^−1^) = Car × Vt × n/FW × 1000.(6)
where FW is the fresh weight (g), Vt is the total volume of the extract (mL), and n is the dilution coefficient.

In addition, for the relative water content (RWC), we measured 1.0 g of fresh leaves (FW) from the NKY298-1 and NKY211 seedlings under all treatments, and then completely immersed them in 50 mL of ddH_2_O until the set weight (TW) was reached. The soaked leaves were then dried in the oven to a set weight (DW). The calculation formula for relative moisture content (RWC) is as follows:RWC = [(FW − DW)/(TW − DW)] × 100%.(7)

For the membrane stability index (MSI), we referred to the method of Zhao et al. [21]. We placed 0.2 g of fresh leaves in a test tube containing 10 mL of ddH_2_O and incubated them at 40 °C for 30 min. The conductivity (C1) was measured using the DDSJ-308F conductivity meter (Rex Electric Chemical, Shanghai, China). They were stored in a 100 °C water bath for 15 min and we recorded the conductivity (C2). The estimated MSI is as follows:MSI = [1 − (C1/C2)] × 100%.(8)

### 4.3. Comprehensive Evaluation Analysis

To analyze the effect of the corresponding concentration of exogenous ZnONPs on the salt tolerance of individual traits of maize under NaCl stress more conveniently, scientifically and accurately, we provided a new definition for the stress improvement index (SII) and alleviation capacity index (ACI) of the corresponding concentration of exogenous ZnONPs application on the individual traits of two maize genotypes under salt stress. The calculation formula for SII as follows:SII_ij(S+ZnONPs-*k*)_ = [*T*_ij(S+ZnONPs-*k*)_ − *T_S-ij_*]/|*T_CK-ij_* − *T_S-ij_*|(9)SII_ij(S+ZnONPs-*k*)_ = [*T_S-ij_* − *T*_ij(S+ZnONPs-*k*)_]/|*T_CK-ij_* − *T_S-ij_*|(10)
where *T*_ij(s+ZnONPs-*k*)_ represented the stress improvement index (SII) of the *j*-th (*j* = 1, 2, 3 … and 26) trait of the *i*-th (*i* = 1 and 2) maize genotype seedlings under the *k*-th (*k* = 10, 50, 100, 150, and 180) concentration of ZnONPs application under NaCl stress. Where *k*-th represented the concentration of ZnONPs of 10, 50, 100, 150, and 180 mM, respectively, *T_ij_*_(*S+ZnONPs-k*)_ was the *i*-th maize genotype seedling treated with S + ZnONPs-*k*. The *T_CK-ij_* represented the value of the *j*-th trait of the two maize genotypes under CK treatment and the *T_S-ij_* represented the value of the *j*-th trait of the two maize genotypes under NaCl stress.

The calculation formula for ACI is as follows:(11)$${ACI}_{ij(S+ZnONPs-k)}=\frac{{|(T}_{ij(S+ZnONPs-k)}-T_{ij\left(S\right)})|}{|T_{ij\left(CK\right)}-T_{ij(S)}|}$$
where *T*_ij(s+ZnONPs-*k*)_ represented the stress improvement index (SII) of the *j*-th (*j* = 1, 2, 3 … and 26) trait of the *i*-th (*i* = 1 and 2) maize genotype seedlings under the *k*-th (*k* = 10, 50, 100, 150, and 180) concentration of ZnONPs application under NaCl stress. Where *k*-th represented the concentration of ZnONPs of 10, 50, 100, 150, and 180 μmol/L, respectively, *T_ij_*_(*S+ZnONPs-k*)_ was the *i*-th maize genotype seedling treated with S + ZnONPs-*k*. The *T_CK-ij_* represented the value of the *j*-th trait of two maize genotypes under CK treatment, and the *T_S-ij_* represented the value of the *j*-th trait of two maize genotypes under NaCl stress.

At the same time, to objectively evaluate the alleviation effect of different concentrations of ZnONPs solutions on salt tolerance traits under NaCl stress, the SII and ACI values of the 26 traits obtained were used as the evaluation indicators for the magnitude of the alleviation effect of different concentrations of exogenous ZnONPs on salt stress in two maize genotypes under salt stress. The membership function method was adopted to comprehensively evaluate the comprehensive improvement effect (CIE) and the comprehensive alleviation ability (CAC) of different concentrations of exogenous ZnONPs on the salt tolerance of maize under salt stress. The CIE and CAC values were calculated as follows:R(G)*_ij_* = (SII*_ij_* − SII_ij(min)_)/(SII_ij(max)_ − SII_ij(min)_)(12)R(G)*_ij_* = 1 − (SII*_ji_* − SII_ij(min)_)/(SII_iji(max)_ − SII_ij(min)_)(13)CIE(G)*_ij_* = 1/m ∑R(G)*_ij_*(14)
where R(G)*_ij_* represented the membership values of the enhanced salt tolerance effect of the corresponding concentration of exogenous ZnONPs under NaCl stress *ij*-th (*i* = 1 and 2, *j* = 1, 2, 3 … and 26), was the treatment with the addition of the *k*-th (*k* = 10, 50, 100, 150, and 180) concentration of exogenous ZnONPs under the salt stress of maize, and SII_ij(s+ZnONPs-*k*)_ represented the SII of the corresponding concentration of exogenous ZnONPs of maize under salt stress. SII_ij(min)_ represented the minimum SII value of a single trait under salt stress conditions treated with all concentrations of exogenous ZnONPs, and SII_ij(max)_ represented the maximum SII value of a single trait under salt stress conditions treated with all concentrations of exogenous ZnONPs.(15)$$U({ACI}_{ij(S+ZnONPs-k)})=\frac{{ACI}_{ij(S+ZnONPs-k)}-{ACI}_{ij(min)}}{{ACI}_{ij(max)}-{ACII}_{ij(min)}}$$(16)$$U({ACI}_{ij(S+ZnONPs-k)})=1-\frac{{ACI}_{ij\left(S+ZnONPs-k\right)}-{ACI}_{ij\left(min\right)}}{{ACI}_{ij\left(max\right)}-{ACI}_{ij\left(min\right)}}$$(17)$${CAC}_{i(S+ZnONPs-k)}=1/m{\sum U(ACI}_{ij(S+ZnONPs-k)})$$
where ${U(ACI}_{ij(S+ZnONPs-k)})$ represented the membership values of the alleviation ability of the corresponding concentration of exogenous ZnONPs under NaCl stress *ij*-th (*i* = 1 and 2, *j* = 1, 2, 3 … and 26) was the treatment with the addition of the *k*-th (*k* = 10, 50, 100, 150, and 180) concentration of exogenous ZnONPs under the salt stress of maize, and ACI_ij(S+ZnONPs-*k)*_ represented the ACI of the corresponding concentration of exogenous ZnONPs of maize under salt stress. ACI_ij(min)_ represented the minimum ACI value of a single trait under salt stress conditions treated with all concentrations of exogenous ZnONPs, and ACI_ij(max)_ represented the maximum ACI value of a single trait under salt stress conditions treated with all concentrations of exogenous ZnONPs. Where S + ZnONPs-k represented the S + ZnONPs-10: 120 mmol/L NaCl solution + 10 μmol/L ZnONPs in 20 °C environment; S + ZnONPs-50: 120 mmol/L NaCl solution + 50 μmol/L ZnONPs in 20 °C environment; S + ZnONPs-100: 120 mmol/L NaCl solution + 100 μmol/L ZnONPs in 20 °C environment; S + ZnONPs-150: 120 mmol/L NaCl solution + 150 μmol/L ZnONPs in 20 °C environment; and S + ZnONPs-180: 120 mmol/L NaCl solution + 180 μmol/L ZnONPs in 20 °C environment.

### 4.4. Statistical Analysis

Analysis of ANOVA for the data of 26 traits under seven treatments was performed using IBM-SPSS Statistics v.20.0 software (SPSS, Chicago, IL, USA; https://www.Ibm.com/products/spss-statistics; viewed on 20 February 2025). The between-group linkage cluster analysis was performed using R package (R version 4.2.3; http://www.R-project.org/, accessed on 3 March 2025).

## Figures and Tables

**Figure 1 plants-14-03104-f001:**
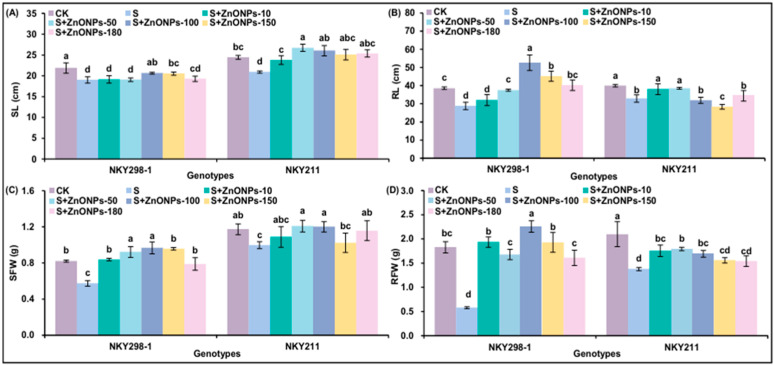
Phenotypic changes in two maize genotypes (NKY298-1 and NKY211) seedlings under different treatments. (**A**). The changes in the seedling length (SL) of two maize genotypes seedlings under different treatments. (**B**). The changes in the root length (RL) of two maize genotypes seedlings under different treatments. (**C**). The changes in the seedling fresh weight (SFW) of two maize genotypes seedlings under different treatments. (**D**). The changes in the root fresh weight (RFW) of two maize genotypes seedlings under different treatments. Lowercase letters represent significance in analysis of ANOVA, *p* < 0.05. CK: 0 mmol/L NaCl solution + 0 μmol/L ZnONPs in 20 °C environment; S: 120 mmol/L NaCl solution in 20 °C environment; S + ZnONPs-10: 120 mmol/L NaCl solution + 10 μmol/L ZnONPs in 20 °C environment; S + ZnONPs-50: 120 mmol/L NaCl solution + 50 μmol/L ZnONPs in 20 °C environment; S + ZnONPs-100: 120 mmol/L NaCl solution + 100 μmol/L ZnONPs in 20 °C environment; S + ZnONPs-150: 120 mmol/L NaCl solution + 150 μmol/L ZnONPs in 20 °C environment; and S + ZnONPs-180: 120 mmol/L NaCl solution + 180 μmol/L ZnONPs in 20 °C environment. The error bars in the manuscript all represent the standard deviations (SD) of the samples under different treatments.

**Figure 2 plants-14-03104-f002:**
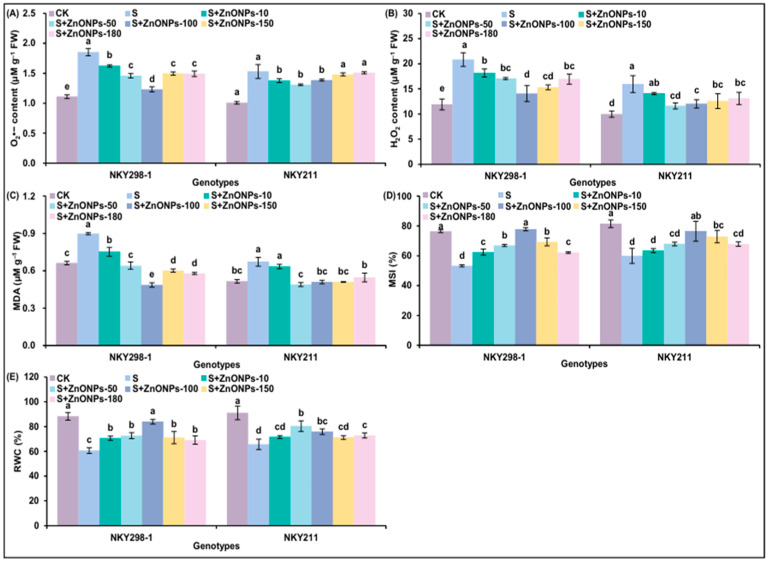
The changes in five traits in seedlings of two maize genotypes under different treatments. CK: 0 mmol/L NaCl solution + 0 μmol/L ZnONPs in 20 °C environment; S: 120 mmol/L NaCl solution in 20 °C environment; S + ZnONPs-10: 120 mmol/L NaCl solution + 10 μmol/L ZnONPs in 20 °C environment; S + ZnONPs-50: 120 mmol/L NaCl solution + 50 μmol/L ZnONPs in 20 °C environment; S + ZnONPs-100: 120 mmol/L NaCl solution + 100 μmol/L ZnONPs in 20 °C environment; S + ZnONPs-150: 120 mmol/L NaCl solution + 150 μmol/L ZnONPs in 20 °C environment; and S + ZnONPs-180: 120 mmol/L NaCl solution + 180 μmol/L ZnONPs in 20 °C environment. (**A**). Superoxide anion content (O_2_^•−^), (**B**). hydrogen peroxide content (H_2_O_2_), (**C**). malondialdehyde content (MDA), (**D**). membrane stability index (MSI), and (**E**). relative water content (RWC). Lowercase letters represent significance in analysis of ANOVA, *p* < 0.05. The contents of H_2_O_2_ (Cat: BC3595) and MDA (Cat: BC6415) are calculated in accordance with the instructions provided on the official website of Solarbio. The calculation method of MSI can be found in Materials and Methods. The error bars in the manuscript all represent the standard deviations (SD) of the samples under different treatments.

**Figure 3 plants-14-03104-f003:**
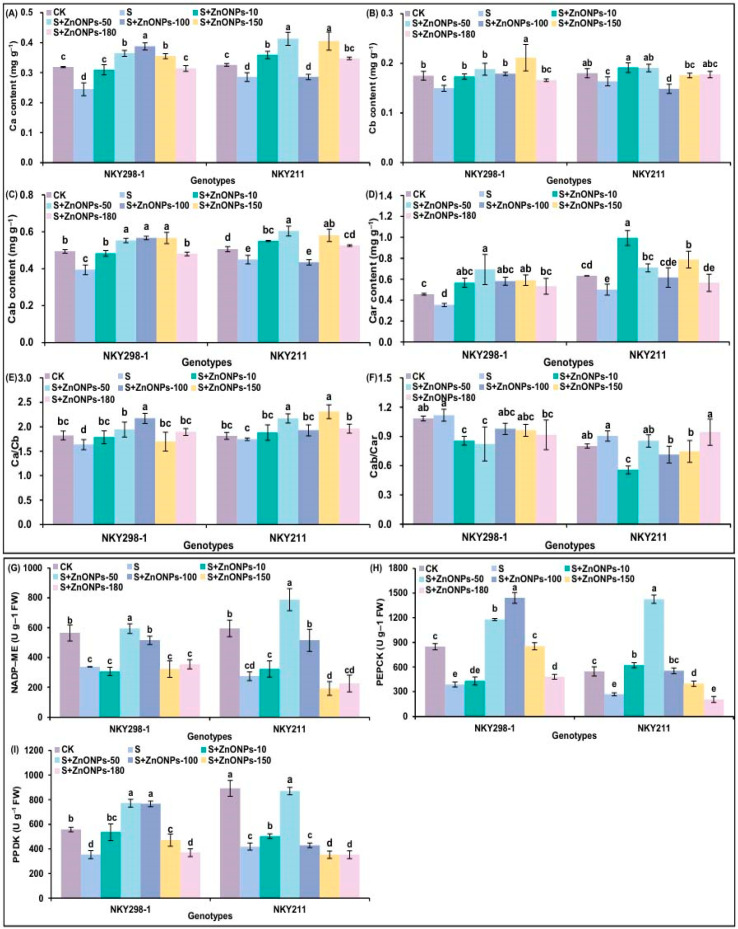
The changes in photosynthetic pigments and photosynthetic enzymes in two maize seedlings under different treatments. CK: 0 mmol/L NaCl solution + 0 μmol/L ZnONPs in 20 °C environment; S: 120 mmol/L NaCl solution in 20 °C environment; S + ZnONPs-10: 120 mmol/L NaCl solution + 10 μmol/L ZnONPs in 20 °C environment; S + ZnONPs-50: 120 mmol/L NaCl solution + 50 μmol/L ZnONPs in 20 °C environment; S + ZnONPs-100: 120 mmol/L NaCl solution + 100 μmol/L ZnONPs in 20 °C environment; S + ZnONPs-150: 120 mmol/L NaCl solution + 150 μmol/L ZnONPs in 20 °C environment; S + ZnONPs-180: 120 mmol/L Nacl solution + 180 μmol/L ZnONPs in 20 °C environment. (**A**). Chlorophyll a content (Ca), (**B**). chlorophyll b content (Cb), (**C**). chlorophyll a + b content (Cab), (**D**). carotenoid content (Car), (**E**). chlorophyll a:b ratio (Ca/Cb), (**F**). chlorophyll a + b content: carotenoid content ratio (Cab/Car), (**G**). NADP-malidase content (NADP-ME), (**H**). phosphoenolpyruvate carboxykinase (PEPCK), and (**I**). pyruvate phosphodikinase content (PPDK). Lowercase letters represent significance in analysis of ANOVA, *p* < 0.05. The error bars in the manuscript all represent the standard deviations (SD) of the samples under different treatments. All the enzyme activities were determined using the Solarbio kit (Solarbio Technology Co., LTD., Beijing, China). The pigment content was determined with 95% ethanol.

**Figure 4 plants-14-03104-f004:**
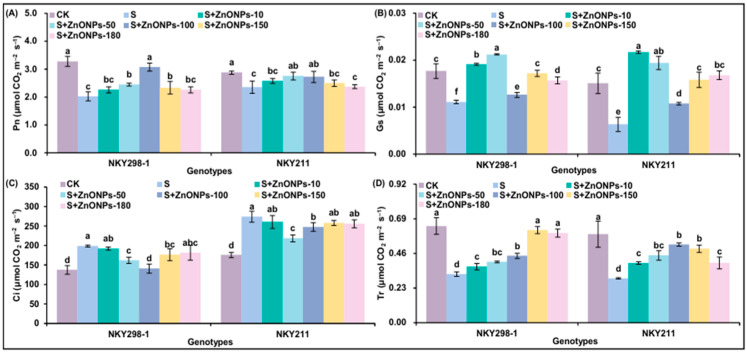
The changes in photosynthetic parameters of seedlings of two maize genotypes under different treatments. CK: 0 mmol/L NaCl solution + 0 μmol/L ZnONPs in 20 °C environment; S: 120 mmol/L NaCl solution in 20 °C environment; S + ZnONPs-10: 120 mmol/L NaCl solution + 10 μmol/L ZnONPs in 20 °C environment; S + ZnONPs-50: 120 mmol/L NaCl solution + 50 μmol/L ZnONPs in 20 °C environment; S + ZnONPs-100: 120 mmol/L NaCl solution + 100 μmol/L ZnONPs in 20 °C environment; S + ZnONPs-150: 120 mmol/L NaCl solution + 150 μmol/L ZnONPs in 20 °C environment; and S + ZnONPs-180: 120 mmol/L NaCl solution + 180 μmol/L ZnONPs in 20 °C environment. (**A**). Net photosynthetic rate (Pn), (**B**). stomatal conductivity (Gs), (**C**). intercellular CO_2_ concentration (Ci), and (**D**). transpiration rate (Tr). Lowercase letters represent significance in analysis of ANOVA, *p* < 0.05. The error bars in the manuscript all represent the standard deviations (SD) of the samples under different treatments.

**Figure 5 plants-14-03104-f005:**
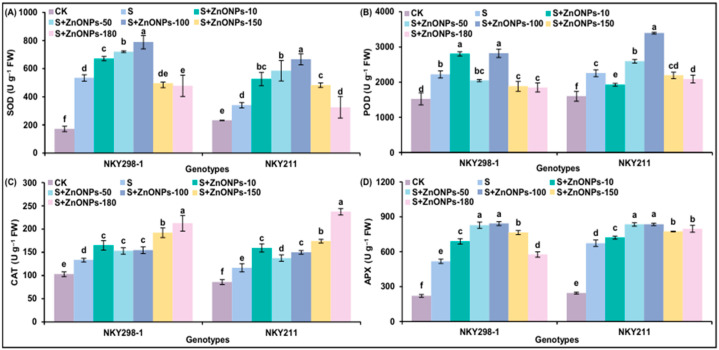
The changes in antioxidant enzymes activity of two maize seedlings under different treatments. CK: 0 mmol/L NaCl solution + 0 μmol/L ZnONPs in 20 °C environment; S: 120 mmol/L NaCl solution in 20 °C environment; S + ZnONPs-10: 120 mmol/L NaCl solution + 10 μmol/L ZnONPs in 20 °C environment; S + ZnONPs-50: 120 mmol/L NaCl solution + 50 μmol/L ZnONPs in 20 °C environment; S + ZnONPs-100: 120 mmol/L NaCl solution + 100 μmol/L ZnONPs in 20 °C environment; S + ZnONPs-150: 120 mmol/L NaCl solution + 150 μmol/L ZnONPs in 20 °C environment; and S + ZnONPs-180: 120 mmol/L Nacl solution + 180 μmol/L ZnONPs in 20 °C environment. (**A**). Superoxide dismutase activity (SOD), (**B**). peroxidase activity (POD), (**C**). catalase activity (CAT), amd (**D**). ascorbate peroxidase activity (APX). Lowercase letters represent significance in analysis of ANOVA, *p* < 0.05. The error bars in the manuscript all represent the standard deviations (SD) of the samples under different treatments. All the enzyme activities were determined using the Solarbio kit.

**Figure 6 plants-14-03104-f006:**
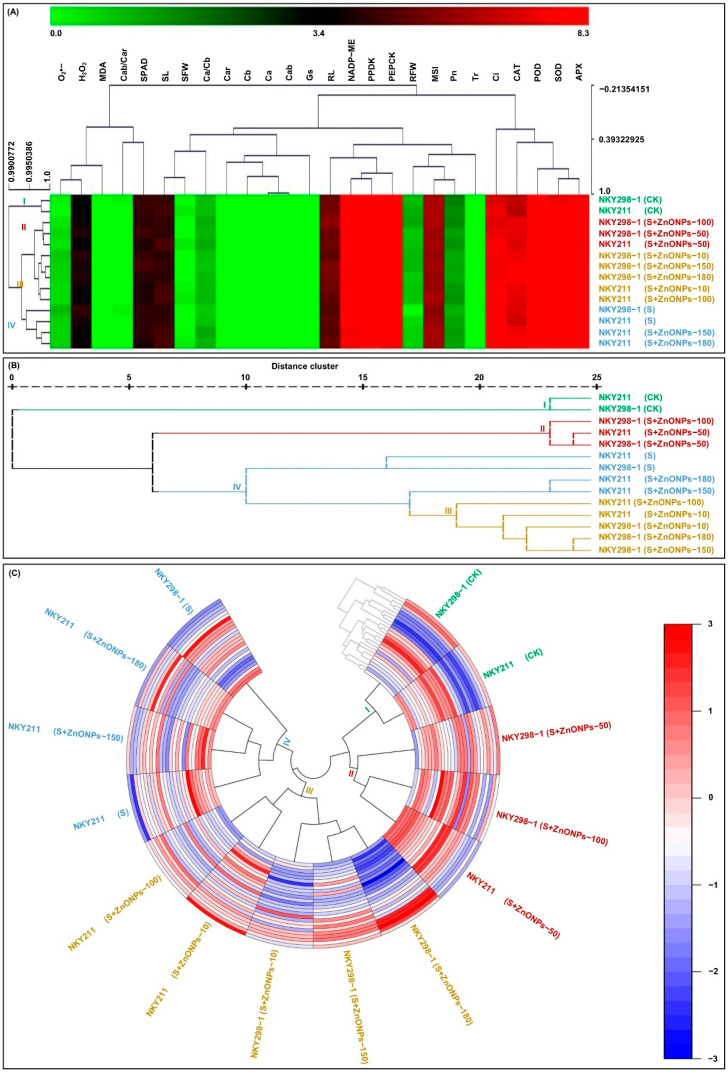
CK: 0 mmol/L NaCl solution + 0 μmol/L ZnONPs in 20 °C environment; S: 120 mmol/L NaCl solution in 20 °C environment; S + ZnONPs-10: 120 mmol/L NaCl solution + 10 μmol/L ZnONPs in 20 °C environment; S + ZnONPs-50: 120 mmol/L NaCl solution + 50 μmol/L ZnONPs in 20 °C environment; S + ZnONPs-100: 120 mmol/L NaCl solution + 100 μmol/L ZnONPs in 20 °C environment; S + ZnONPs-150: 120 mmol/L NaCl solution + 150 μmol/L ZnONPs in 20 °C environment; and S + ZnONPs-180: 120 mmol/L NaCl solution + 180 μmol/L ZnONPs in 20 °C environment. (**A**). Hierarchical clustering heatmap analysis of 26 traits of two maize genotypes under different treatments. (**B**). Intergroup linkage clustering analysis of two maize genotypes under different treatments. (**C**). Analysis of circular clustering heat maps of two maize genotypes under different treatments. I, II, III, and IV represent clustering groups.

**Figure 7 plants-14-03104-f007:**
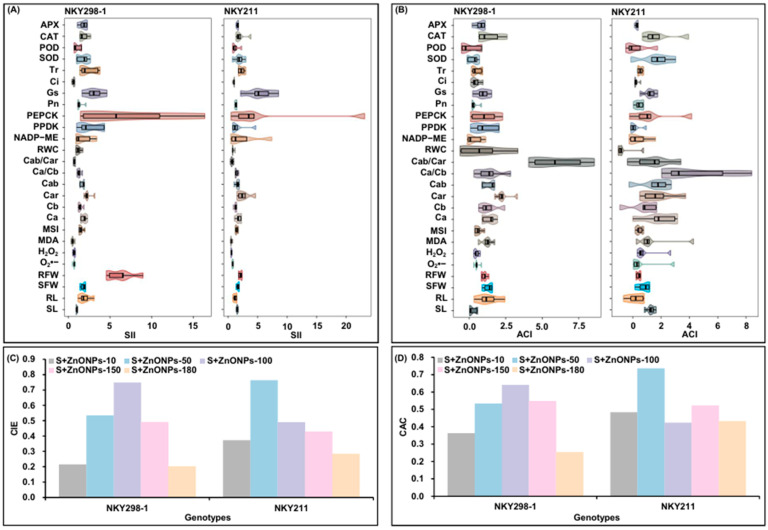
The alleviation effects of 26 traits of two maize genotypes under the addition of ZnONPs and the alleviation effects of different concentrations of ZnONPs on NaCl stress were statistically analyzed by using the membership function method. Traits included ascorbate peroxidase activity (APX), catalase activity (CAT), peroxidase activity (POD), superoxide dismutase activity (SOD), transpiration rate (Tr), intercellular CO_2_ concentration (Ci), stomatal conductance (Gs), net photosynthetic rate (Pn), phosphoenolpyruvate carboxykinase (PEPCK), pyruvate phosphodikinase content (PPDK), NADP-malidase content (NADP-ME), soil–plant analysis development (SPAD), chlorophyll a + b content, carotenoid content ratio (Cab/Car), chlorophyll a:b ratio (Ca/Cb), chlorophyll a + b content (Cab), carotenoid content (Car), chlorophyll b content (Cb), chlorophyll a content (Ca), membrane stability index (MSI), malondialdehyde content (MDA), H_2_O_2_ content (H_2_O_2_), O_2_^•−^ content (O_2_^•−^), root fresh weight (RFW), seedling fresh weight (SFW), root length (RL), and seedling length (SL). CK: 0 mmol/L NaCl solution + 0 μmol/L ZnONPs in 20 °C environment; S: 120 mmol/L NaCl solution in 20 °C environment; S + ZnONPs-10: 120 mmol/L NaCl solution + 10 μmol/L ZnONPs in 20 °C environment; S + ZnONPs-50: 120 mmol/L NaCl solution + 50 μmol/L ZnONPs in 20 °C environment; S + ZnONPs-100: 120 mmol/L NaCl solution + 100 μmol/L ZnONPs in 20 °C environment; S + ZnONPs-150: 120 mmol/L NaCl solution + 150 μmol/L ZnONPs in 20 °C environment; and S + ZnONPs-180: 120 mmol/L NaCl solution + 180 μmol/L ZnONPs in 20 °C environment. (**A**). Stress improvement index (SII) among 26 traits of five concentrations of ZnONPs solutions on two maize genotypes under NaCl stress. (**B**). Alleviation capacity index (ACI) among 26 traits of five concentrations of ZnONPs solutions on two maize genotypes under NaCl stress. (**C**). Comprehensive improvement effects (CIE) of five concentrations of ZnONPs solutions on two maize genotypes under NaCl stress. (**D**). Comprehensive alleviation capacity (CAC) of five concentrations of ZnONPs solutions on two maize genotypes under NaCl stress.

**Table 1 plants-14-03104-t001:** Factor ANOVA of 26 traits in two maize genotype seedlings under different treatments (maize genotype and Na, Zn).

Variation Source	Genotype (G)	NaCl Treatments (Na)	ZnONPs Treatments (Zn)	G*Na Interaction	G*Zn Interaction
SL	F = 238.924 ***	F = 43.881 ***	F = 13.281 ***	F = 0.475	F = 7.928 ***
RL	F = 30.596 ***	F = 39.334 ***	F = 15.619 ***	F = 1.028	F = 35.111 ***
SFW	F = 120.192 ***	F = 29.595 ***	F = 14.817 ***	F = 0.838	F = 5.075 ***
RFW	F = 2.973	F = 184.176 ***	F = 46.861 ***	F = 13.256 **	F = 21.488 ***
O_2_^−^	F = 13.553 **	F = 619.889 ***	F = 51.973 ***	F = 18.392 ***	F = 24.471 ***
H_2_O_2_	F = 61.992 ***	F = 151.111 ***	F = 19.576 ***	F = 5.916 *	F = 2.244
MDA	F = 144.284 ***	F = 256.253 ***	F = 152.402 ***	F = 10.394 **	F = 25.714 ***
MSI	F = 8.361 **	F = 184.790 ***	F = 37.383 ***	F = 0.228	F = 1.839
Chl a	F = 5.365 *	F = 45.360 ***	F = 54.054 ***	F = 3.877	F = 25.267 ***
Chl b	F = 0.836	F = 12.326 **	F = 12.089 ***	F = 0.554	F = 7.824 ***
Car	F = 37.387 ***	F = 9.077 **	F = 20.926 ***	F = 0.144	F = 8.105 ***
Chl a + b	F = 1.657	F = 50.035 ***	F = 54.181 ***	F = 3.670	F = 23.215 ***
Chl a/b	F = 5.289 *	F = 3.560	F = 8.887 ***	F = 0.809	F = 8.380 ***
Cab/Car	F = 31.818 ***	F = 1.722	F = 7.644 ***	F = 0.426	F = 3.975 **
RWC	F = 1.470	F = 208.415 ***	F = 19.345 ***	F = 0.380	F = 4.451 **
NADP-ME	F = 0.075	F = 96.247 ***	F = 75.305 ***	F = 2.802	F = 9.410 ***
PPDK	F = 13.716 **	F = 249.301 ***	F = 131.605 ***	F = 39.676 ***	F = 26.825 ***
PEPCK	F = 309.647 ***	F = 254.963 ***	F = 563.513 ***	F = 15.325 **	F = 168.660 ***
Pn	F = 1.342	F = 112.291 ***	F = 17.573 ***	F = 19.124 ***	F = 4.838 **
Gs	F = 4.235 *	F = 117.556 ***	F = 87.524 ***	F = 2.339	F = 6.670 ***
Ci	F = 239.532 ***	F = 148.079 ***	F = 15.226 ***	F = 8.028 **	F = 3.293 *
Tr	F = 11.146 **	F = 242.406 ***	F = 39.524 ***	F = 0.422	F = 14.484 ***
SOD	F = 21.755 ***	F = 162.378 ***	F = 97.001 ***	F = 47.710 ***	F = 5.457 **
POD	F = 12.210 **	F = 126.359 ***	F = 92.638 ***	F = 0.111	F = 40.249 ***
CAT	F = 6.578 *	F = 40.942 ***	F = 106.514 ***	F = 0.001	F = 5.732 **
APX	F = 34.835 ***	F = 1111.396 ***	F = 148.302 ***	F = 36.445 ***	F = 38.032 ***

* represents *p* < 0.05, ** represents *p* < 0.01, *** represents *p* < 0.001.

**Table 2 plants-14-03104-t002:** Treatment timeline schematic.

Treatments	NaCl Concentrations (mmol/L)	ZnONPs Concentrations (μmol/L)
CK	0	0
S	120	0
S + ZnONPs-10	120	10
S + ZnONPs-50	120	50
S + ZnONPs-100	120	100
S + ZnONPs-150	120	150
S + ZnONPs-180	120	180

## Data Availability

Not applicable.

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
