# Peer review of "ZnONPs Alleviates Salt Stress in Maize Seedlings by Improving Antioxidant Defense and Photosynthesis Potential"

_plants, 2025, doi:10.3390/plants14193104_

Round 1
Reviewer 1 Report
Comments and Suggestions for Authors
Suggestions for Improvement
This study investigates the effect of foliar and seed exposure to zinc-oxide nanoparticles (ZnONPs) on reducing NaCl-induced salt stress in two maize inbred lines (NKY298-1 and NKY211). Over a period of seven days under 120 mM NaCl, the authors measure growth, reactive oxygen species (ROS)/malondialdehyde (MDA), membrane stability index (MSI), water status (relative water content, RWC), photosynthetic pigments and parameters, selected photosynthetic enzymes, and antioxidant enzyme activities. They propose four composite indices (SII, ACI, CIE, CAC) to rank the effectiveness of ZnONP doses. The findings suggest that 50–100 μM ZnONPs significantly mitigate salt damage, with optimal doses varying by genotype (50 μM for NKY211 and 100 μM for NKY298-1). Notably, the study's originality lies in its side-by-side genotype comparison and composite scoring across 26 traits. However, the interpretation of mechanisms is constrained by a lack of nanomaterial characterization and other methodological issues.
Major Concerns
Nanomaterial Characterization: The study lacks characterization of the nanomaterial. Without data on size, polydispersity index (PDI), zeta potential, and morphology, it is unclear what the seedlings were exposed to. Molar dosing for nanoparticles is non-standard. Please, provide a certificate of analysis from the vendor and minimal in-house characterization (e.g., dynamic light scattering (DLS) size/PDI and zeta potential in distilled water and 120 mM NaCl, plus transmission electron microscopy (TEM) images). Report mass concentrations alongside μM estimates and discuss stability in saline conditions.
- Missing Ionic-Zinc Control: Without a control using ZnSO₄, it is challenging to determine whether the observed benefits are specific to nanoparticles or due to ionic zinc. Please introduce controls for core measurements: (i) 120 mM NaCl + ZnSO₄ at equivalent zinc concentrations to ZnONPs, and (ii) ZnONPs without salt. Present these findings in a supplementary figure.
- Ambiguous Exposure Design: The exposure method is unclear, and the selection of seedlings introduces bias.Please clarify the exposure sequence and specify whether the sample size refers to pots or individual plants. Re-analyze using randomly selected seedlings or clearly describe the selection criteria.
- Units and Notation Errors: There are inconsistencies in gas exchange units and enzyme terminology. Please standardize units to μmol CO₂ m⁻² s⁻¹ for photosynthesis, and correct enzyme names and activity reporting.
- Statistics and Multiple Testing: The statistical analysis lacks detail regarding assumptions and post-hoc tests. Please specify the statistical methods used, report effectsizes, and address multiple testing corrections across traits.
- Composite Indices Verification: The equations for the composite indices need verification for accuracy and clarity. Please move detailed derivations to supplementary materials and provide a table mapping trait directions.
- Language and Clarity: Grammatical errors and spelling mistakes impede readability. Please engage professional editing services for language refinement.
Minor Comments
Title/Abstract: Ensure correct grammar in the title. Clarify percentages in the abstract and define terms like MSI and CIE/CAC.
Introduction: Avoid overgeneralizations and provide context. Cite relevant literature and frame objectives as testable hypotheses.
Results-Fig.1: Include sample size per bar and confirm data representation.
Results-Fig.2: Specify methods for ROS/MDA and MSI calculations.
Results-Figs.3–4: Ensure correct enzyme naming and consistent temperature reporting.
Results-Fig.5: Verify reported changes in enzyme activities and provide applicable units.
Results-Figs.6–8: Provide significance markers for correlations and clarify clustering methods.
Methods: Define experimental units and randomization procedures. Ensure all gas exchange units are correct.
Authorship/Funding: Remove stray characters from the declaration.
References: Standardize formatting and ensure completeness.
Methods & Reproducibility
Replicates: Clearly state biological replicates and avoid biased selection.
Identifiers: Provide vendor information for ZnONPs and complete catalog numbers for kits.
Instrument Settings: Document full leaf chamber settings and stabilization times.
Protocols: Cite standard methods for measurements.
Graphs: Clearly define error bars and include sample sizes in figures.
Figures & Tables
Units: Correct all units and ensure consistency.
Legends: Define abbreviations and indicate statistical methods used.
Scale Bars: Add to any image panels included.
Figure Schematic: Include a diagram of the treatment timeline.
Writing & Formatting
Conduct a thorough editing to correct grammar and terminology throughout the manuscript. Ensure consistency in notation and species formatting.
Comments on the Quality of English Language
Enhance the clarity, coherence, and flow of the narration while improving the readability, fluency, and overall quality of the writing.
Author Response
Dear Editor and Reviewers
Thank you for your letter of – and for the referee’s comments concerning our manuscript, “ZnONPs Alleviates Salt Stress in Maize Seedlings by Improving Antioxidant Defense and Photosynthesis Potential (Manuscript ID: plants-3867181)”. We have carefully studied these comments and have made corresponding corrections to the manuscript, which we describe in detail below. We would like to re-submit the manuscript and that for possible publication on the Special Issue: “Mitigation Strategies and Tolerance of Plants to Abiotic Stresses—2nd Edition” of Plants. Thank you very much for your time and consideration.
Reviewer 1
- Suggestions for Improvement: “This study investigates the effect of foliar and seed exposure to zinc-oxide nanoparticles (ZnONPs) on reducing NaCl-induced salt stress in two maize inbred lines (NKY298-1 and NKY211). Over a period of seven days under 120 mM NaCl, the authors measure growth, reactive oxygen species (ROS)/malondialdehyde (MDA), membrane stability index (MSI), water status (relative water content, RWC), photosynthetic pigments and parameters, selected photosynthetic enzymes, and antioxidant enzyme activities. They propose four composite indices (SII, ACI, CIE, CAC) to rank the effectiveness of ZnONP doses. The findings suggest that 50–100 μM ZnONPs significantly mitigate salt damage, with optimal doses varying by genotype (50 μM for NKY211 and 100 μM for NKY298-1). Notably, the study's originality lies in its side-by-side genotype comparison and composite scoring across 26 traits. However, the interpretation of mechanisms is constrained by a lack of nanomaterial characterization and other methodological issues.”
Thanks for your comments. We agree with your suggestion fully, and we have revised the relevant content in the manuscript in lines 14-25. We then have re-submitted the manuscript.
Thank you for your consideration.
- Nanomaterial Characterization: The study lacks characterization of the nanomaterial. Without data on size, polydispersity index (PDI), zeta potential, and morphology, it is unclear what the seedlings were exposed to. Molar dosing for nanoparticles is non-standard. Please, provide a certificate of analysis from the vendor and minimal in-house characterization (e.g., dynamic light scattering (DLS) size/PDI and zeta potential in distilled water and 120 mM NaCl, plus transmission electron microscopy (TEM) images). Report mass concentrations alongside μM estimates and discuss stability in saline conditions.
Thanks for your comments. We purchased material ZnONPs from the official website of Yuan Ye (https://www.shyuanye.com/goods-V33683.html), the product information is shown in the following picture. We are very sorry that the supplier did not provide us with the corresponding characterization and other data. Meanwhile, due to a series of issues such as funds and equipment, we have not conducted a further in-depth analysis of its structure. However, if there is an opportunity, we would be very willing to further explore and study it. We then have re-submitted the manuscript. Thank you very much!
Thank you for your consideration.
- Missing Ionic-Zinc Control: Without a control using ZnSO₄, it is challenging to determine whether the observed benefits are specific to nanoparticles or due to ionic zinc. Please introduce controls for core measurements: (i) 120 mM NaCl + ZnSO₄ at equivalent zinc concentrations to ZnONPs, and (ii) ZnONPs without salt. Present these findings in a supplementary figure.
Thanks for your comments. We fully agree with your suggestions. At the same time, the previous study showed that “Both ZnO SMPs and ZnO NPs, in the concentration range from 50 to 1600 mg∙L−1, can be used to stimulate the germination process of onion seeds, without negative effects on the further growth and development of seedlings. There were no differences found between the action of ZnO NPs and ZnO SMPs, which suggested that the most important factor influencing seed germination was in fact the concentration of zinc ions, not the particle size.” (Tymoszuk et al. Zinc Oxide and Zinc Oxide Nanoparticles Impact on In Vitro Germination and Seedling Growth in Allium cepa L. Materials, 2020, 13). Therefore, we did not conduct ion control experiments, but we will incorporate your suggestions into our subsequent research. We then have re-submitted the manuscript.
Thank you for your consideration.
- Ambiguous Exposure Design: The exposure method is unclear, and the selection of seedlings introduces bias. Please clarify the exposure sequence and specify whether the sample size refers to pots or individual plants. Re-analyze using randomly selected seedlings or clearly describe the selection criteria.
Thanks for your comments. In this study, first, we soaked the seeds in the corresponding solution for 24 hours, then sowed them into the corresponding culture medium. After 7 days of cultivation, we selected three seedlings with the same growth from each pot under different treatments for measurement and the sample size refers to the size of a single plant. We have further improved the corresponding content in section 4.2. Phenotypic and Physiological Assays. We then have re-submitted the manuscript.
Thank you for your consideration.
- Units and Notation Errors: There are inconsistencies in gas exchange units and enzyme terminology. Please standardize units to μmol CO₂ m⁻² s⁻¹ for photosynthesis, and correct enzyme names and activity reporting.
Thanks for your comments. As suggested, we have checked the relevant content in the manuscript and corrected it in the corresponding chapters. We then have re-submitted the manuscript.
Thank you for your consideration.
- Statistics and Multiple Testing: The statistical analysis lacks detail regarding assumptions and post-hoc tests. Please specify the statistical methods used, report effect sizes, and address multiple testing corrections across traits.
Thanks for your comments. As suggestion, we have improved the relevant content in the manuscript in the Materials and methods section. We then have re-submitted the manuscript.
Thank you for your consideration.
- Composite Indices Verification: The equations for the composite indices need verification for accuracy and clarity. Please move detailed derivations to supplementary materials and provide a table mapping trait directions.
Thanks for your comments. As suggested, we once again verified the correctness of the formulas and listed all of them in the supplementary materials. We then have re-submitted the manuscript.
Thank you for your consideration.
- Language and Clarity: Grammatical errors and spelling mistakes impede readability. Please engage professional editing services for language refinement.
Thanks for your comments. During the revision process, we also jointly modified the language and readability of the manuscript. In the subsequent manuscript writing process, we will pay more attention to the language issue. I'm very sorry for the trouble caused to you. We then have re-submitted the manuscript.
Thank you for your consideration.
- Title/Abstract: Ensure correct grammar in the title. Clarify percentages in the abstract and define terms like MSI and CIE/CAC!
Thanks for your comments. As suggested, we have checked and revised the relevant content in title and abstract. We then have re-submitted the manuscript.
Thank you for your consideration.
- Introduction: Avoid overgeneralizations and provide context. Cite relevant literature and frame objectives as testable hypotheses.
Thanks for your comments. As suggested, we have checked and revised the relevant content and modified the literature citations in the introduction section. We then have re-submitted the manuscript.
Thank you for your consideration.
- Results-Fig.1: Include sample size per bar and confirm data representation.
Thanks for your comments. As suggested, we have once again confirmed that the data representation is correct. We then have re-submitted the manuscript.
Thank you for your consideration.
- Results-Fig.2: Specify methods for ROS/MDA and MSI calculations.
Thanks for your comments. The calculation methods for the contents of the three substances have all been presented in the text, and the relevant contents have been further improved in lines 164-167. We then have re-submitted the manuscript.
Thank you for your consideration.
- Results-Figs.3–4: Ensure correct enzyme naming and consistent temperature reporting.
Thanks for your comments. As suggested, we rechecked and confirmed the correctness of the enzyme names and consistent temperature reporting shown in the manuscript. We then have re-submitted the manuscript.
Thank you for your consideration.
- Results-Fig.5: Verify reported changes in enzyme activities and provide applicable units.
Thanks for your comments. As suggested, we verified the changes in enzyme activity in the manuscript and checked the units used. We then have re-submitted the manuscript.
Thank you for your consideration.
- Results-Figs.6–8: Provide significance markers for correlations and clarify clustering methods.
Thanks for your comments. Significance analysis markers have been added to the correlation analysis. Three clustering methods were used in the manuscript, namely hierarchical clustering heatmap analysis (Figure 6A), inter-cluster linkage clustering analysis (Figure 6B), and circular clustering heatmap analysis (Figure 6C), all of which have been explained in the figure captions. We then have re-submitted the manuscript.
Thank you for your consideration.
- Methods: Define experimental units and randomization procedures. Ensure all gas exchange units are correct.
Thanks for your comments. As suggested, we have checked the issues you pointed out and ensured they are correct. We then have re-submitted the manuscript.
Thank you for your consideration.
- Authorship/Funding: Remove stray characters from the declaration.
Thanks for your comments. As suggested, we have checked and revised the relevant content. We then have re-submitted the manuscript.
Thank you for your consideration.
- References: Standardize formatting and ensure completeness.
Thanks for your comments. As suggested, we have checked and improved the relevant content. We then have re-submitted the manuscript.
Thank you for your consideration.
- Replicates: Clearly state biological replicates and avoid biased selection.
Thanks for your comments. As suggested, we have already revised the relevant content in the Materials and Methods section. We then have re-submitted the manuscript.
Thank you for your consideration.
- Identifiers: Provide vendor information for ZnONPs and complete catalog numbers for kits.
Thanks for your comments. As suggested, we have refined and added the relevant content in the 4.1. Maize Materials and Experimental Design section. We then have re-submitted the manuscript.
Thank you for your consideration.
- Instrument Settings: Document full leaf chamber settings and stabilization times.
Thanks for your comments. The relevant content has been presented in the manuscript 4.1. Maize Materials and Experimental Design section. We then have re-submitted the manuscript.
Thank you for your consideration.
- Protocols: Cite standard methods for measurements
Thanks for your comments. The measurement methods adopted in the research are all standard measurement methods that the team has learned over the years. However, we will further improve the relevant measurement methods and approaches based on your suggestions. We then have re-submitted the manuscript.
Thank you for your consideration.
- Graphs: Clearly define error bars and include sample sizes in figures.
Thanks for your comments. As suggested, we have clearly defined the error bars in the manuscript and added them to the corresponding captions. We then have re-submitted the manuscript.
Thank you for your consideration.
- Units: Correct all units and ensure consistency.
Thanks for your comments. As suggested, we have checked all the units in the manuscript and corrected the corresponding incorrect units, and ensured their consistency. We then have re-submitted the manuscript.
Thank you for your consideration.
- Legends: Define abbreviations and indicate statistical methods used.
Thanks for your comments. As suggested, we have refined and revised the corresponding content. We then have re-submitted the manuscript.
Thank you for your consideration.
- Scale Bars: Add to any image panels included.
Thank you very much for your suggestion. During the subsequent manuscript writing process, we will definitely add the scale to the corresponding position. We then have re-submitted the manuscript.
Thank you for your consideration.
- Figure Schematic: Include a diagram of the treatment timeline.
Thank you very much for your suggestion. The relevant experimental content has already been presented in the Materials and Methods. However, during the subsequent manuscript writing process, we will definitely refer to your suggestion and list the specific timeline so as to be able to present our experimental process in more depth. We then have re-submitted the manuscript.
Thank you for your consideration.
- Conduct a thorough editing to correct grammar and terminology throughout the manuscript. Ensure consistency in notation and species formatting.
Thanks for your comments. As suggested, we have inspected and corrected the entire manuscript and ensured the consistency of its content. We then have re-submitted the manuscript.
Thank you for your consideration.
- Enhance the clarity, coherence, and flow of the narration while improving the readability, fluency, and overall quality of the writing.
Thanks for your comments. As suggested, we have checked and revised the language part of the entire manuscript to improve the overall quality of the manuscript as much as possible. We are very sorry for the inconvenience caused to you. We then have re-submitted the manuscript.
Thank you for your consideration.
Best wishes!
Xiaoqiang Zhao Professor
State Key Laboratory of Aridland Crop Science, Gansu Agricultural University
- mail: zhaoxiaoq@gsau.edu.cn

Reviewer 2 Report
Comments and Suggestions for Authors
The manuscript by Sun et al. aims to find optimal concentration of ZNO nanoparticles to counteract of salt stress. The study has several problems, that should be addressed before consideration for publication.
First, authors claim they are using ZnO nanoparticles, however the manufacturer information contradict it. The company sells ZnO, not ZnO nanoparticles. How the authors obtained actual ZnO? What was the size of nanoparticles? What was their surface properties/surface stabilization? These parameters are crucial for any nanoparticles application. How authors determined nanoparticles concentration in umols? Was it concentration of ZnO in total or nanoparticles (both values will be different, as ZnO nanoparticles will contains tens of Zn and O atoms). What is also important, the Authors should check if their ZnO nanoparticles sustain in soil - if not, this study is not about nanoparticles, but about not defined product of ZnO reaction with soil compounds.
Introduction misses short paragraph, describing what are nanoparticles and what differs them from bulk materials of the same composition.
It is not clear, why 120 mM NaCl concentration was used to induce salt stress, and why used maize genotypes were selected.
Correlation analysis doesn’t make any sense, or is wrongly described. It is impossible to correlate a change at one point (simply, there is no change in parameters to correlate).
Discussion should include nanoparticles properties and proposal of the mechanism of observed alteration in plants growth.
Figures quality should be improved - some of the colors on the bar graphs legend are hard to be distinguished both in print and on the screen.
Language is a serious problem of this paper. There are several sentences with wrong grammar form of words, wrong sentence order, etc. The manuscript should be drastically revised by English native speaker.
Author Response
Dear Editor and Reviewers
Thank you for your letter of – and for the referee’s comments concerning our manuscript, “ZnONPs Alleviates Salt Stress in Maize Seedlings by Improving Antioxidant Defense and Photosynthesis Potential (Manuscript ID: plants-3867181)”. We have carefully studied these comments and have made corresponding corrections to the manuscript, which we describe in detail below. We would like to re-submit the manuscript and that for possible publication on the Special Issue: “Mitigation Strategies and Tolerance of Plants to Abiotic Stresses—2nd Edition” of Plants. Thank you very much for your time and consideration.
Reviewer 2
- First, authors claim they are using ZnO nanoparticles, however the manufacturer information contradict it. The company sells ZnO, not ZnO nanoparticles. How the authors obtained actual ZnO? What was the size of nanoparticles? What was their surface properties/surface stabilization? These parameters are crucial for any nanoparticles application. How authors determined nanoparticles concentration in umols? Was it concentration of ZnO in total or nanoparticles (both values will be different, as ZnO nanoparticles will contains tens of Zn and O atoms). What is also important, the Authors should check if their ZnO nanoparticles sustain in soil - if not, this study is not about nanoparticles, but about not defined product of ZnO reaction with soil compounds
Thanks for your comments. The specific product information of the nano zinc oxide we use is shown in this picture. Previous studies have shown that “Both ZnO SMPs and ZnO NPs, in the concentration range from 50 to 1600 mg∙L−1, can be used to stimulate the germination process of onion seeds, without negative effects on the further growth and development of seedlings. There were no differences found between the action of ZnO NPs and ZnO SMPs, which suggested that the most important factor influencing seed germination was in fact the concentration of zinc ions, not the particle size.” (Tymoszuk et al. Zinc Oxide and Zinc Oxide Nanoparticles Impact on In Vitro Germination and Seedling Growth in Allium cepa L. Materials, 2020, 13). Therefore, the set concentration refers to the total concentration of ZnONPs. Besides, the soil used in the experiment was not the cultivated soil in the field, but treated vermiculite was used as the culture substrate. We then have re-submitted the manuscript.
Thank you for your consideration.
- Introduction misses short paragraph, describing what are nanoparticles and what differs them from bulk materials of the same composition
Thanks for your comments. We fully agree with your opinion. As suggested, we have already added the relevant content of the comparison between the two in lines 83-85. We then have re-submitted the manuscript.
Thank you for your consideration.
- It is not clear, why 120 mM NaCl concentration was used to induce salt stress, and why used maize genotypes were selected.
Thanks for your comments. The selection of salt concentration was based on the data obtained from the team's previous research, and after multiple experiments and considerations, the two maize genotypes in the manuscript were chosen. We then have re-submitted the manuscript.
Thank you for your consideration.
- Correlation analysis doesn’t make any sense, or is wrongly described. It is impossible to correlate a change at one point (simply, there is no change in parameters to correlate).
Thanks for your comments. We fully agree with your suggestion, and the relevant content was deleted. We then have re-submitted the manuscript.
Thank you for your consideration.
- Discussion should include nanoparticles properties and proposal of the mechanism of observed alteration in plants growth
Thanks for your comments. We fully agree with your opinion. As suggested, the relevant content has already been presented in the discussion section. We then have re-submitted the manuscript.
Thank you for your consideration.
- Figures quality should be improved - some of the colors on the bar graphs legend are hard to be distinguished both in print and on the screen
Thanks for your comments. As suggested, we have adjusted the color of the bar chart in the manuscript. We then have re-submitted the manuscript.
Thank you for your consideration.
- Language is a serious problem of this paper. There are several sentences with wrong grammar form of words, wrong sentence order, etc. The manuscript should be drastically revised by English native speaker.
Thanks for your comments. As suggested, we have checked and revised the language part of the entire manuscript to improve the overall quality of the manuscript as much as possible. We are very sorry for the inconvenience caused to you. We then have re-submitted the manuscript.
Thank you for your consideration.
Best wishes!
Xiaoqiang Zhao Professor
State Key Laboratory of Aridland Crop Science, Gansu Agricultural University
- mail: zhaoxiaoq@gsau.edu.cn

Reviewer 3 Report
Comments and Suggestions for Authors
This paper entitled “ZnONPs Alleviates Salt Stress in Maize Seedlings by Improving Antioxidant Defense and Photosynthesis Potential” This study aimed to investigate the effects of ZnONPs on maize under salt stress. To explore this, two maize genotypes, NKY298-1 and NKY211, were analyzed for changes in phenotype, reactive oxygen species balance, membrane system condition, antioxidant activity, and photosynthetic capacity with different ZnONP applications under 120 mmol/L NaCl stress. The results showed that 120 mmol/L NaCl stress induced oxidative stress in both maize seedlings, with malondialdehyde (MDA), hydrogen peroxide (H2O2), and superoxide anion (O2•−) levels increasing by 24.81%, 67.27%, and 59.32%, respectively. This stress also affected membrane stability, reducing the membrane stability index (MSI) by 28.38%, and lowered the net photosynthetic rate (Pn) by 25% compared with the CK treatment. Importantly, the alleviating effects of various ZnONP concentrations differed among maize genotypes under NaCl stress.
This objective is highly relevant, given the increasing importance of developing salt-tolerant crops in the context of global soil salinization.
The authors used appropriate methodology and experimental design to test the study hypothesis, also.
I consider this topic to be relevant to the plants research field. It is an important work that could be helpful to researchers and appealing to readers of Plants.
The paper is well-organized, easily readable, and presented in a well-structured manner. The figures and tables are appropriate and easy to understand.
The bibliography used is well-documented, and the references are appropriate for the paper.
Therefore, I recommend that the authors address the following aspects to enhance the quality of their study.
- Abstract
The sentences are sometimes awkward and redundant.
It is not clear how ZnONPs are applied (in nutrient solutions? foliar treatment? exact concentrations tested).
The number of concentrations tested was not specified, nor was it clear why two different genotypes were used.
- Introduction
Long sentences, cumbersome structure, repetitions
Please explain why you used a concentration of 120mM NaCl and why you used two genotypes for corn. Justify your answer.
Please state the study hypothesis explicitly.
You state that "maize is affected by salt levels of more than 250 mM NaCl" but you are experimenting with 120 mM NaCl. The choice of this concentration needs to be explained (e.g., sublethal but strong enough to induce stress).
"Causes a 65% yield loss" – this statement is very general; the context should be specified (at what salinity level? Under what conditions?).
It is mentioned that “ZnONPs have been used successfully on other species (cucumber, rapeseed, sorghum, lupin), but not on corn”. However, the authors should specify why corn is a relevant model (e.g., strategic crop, sensitive to salinity, lack of studies with ZnONPs).
- Discussion
For the most part, the discussion reiterates what you have already stated in the Results section, with minor comparisons to the literature.
A mechanistic interpretation is missing (why do ZnONPs decrease ROS? Is it due to catalytic action, role in Zn²⁺ homeostasis, or activation of zinc-dependent antioxidant enzymes?).
The possibility that the effects of ZnONPs depend on the size, shape, and loading of the nanoparticles has not been discussed.
It is stated that "ZnONPs alleviated oxidative stress" without discussing whether the effect is direct (interaction with ROS) or indirect (through increased antioxidant enzyme activity, osmotic or hormonal regulation).
- Please specify in the text from the first appearance, what each abbreviated notation represents.
- The manuscript contains long and redundant sentences, and the phrasing is difficult to follow.
- The authors should emphasize the novelty of their work.
- The limitations of this study, challenges and future perspectives should be included in the manuscript.
- The authors should add a graphical abstract that highlights their work to a broader audience.
Comments on the Quality of English Language
Long sentences, cumbersome structure, repetitions.
The sentences are sometimes awkward and redundant.
Author Response
Dear Editor and Reviewers
Thank you for your letter of – and for the referee’s comments concerning our manuscript, “ZnONPs Alleviates Salt Stress in Maize Seedlings by Improving Antioxidant Defense and Photosynthesis Potential (Manuscript ID: plants-3867181)”. We have carefully studied these comments and have made corresponding corrections to the manuscript, which we describe in detail below. We would like to re-submit the manuscript and that for possible publication on the Special Issue: “Mitigation Strategies and Tolerance of Plants to Abiotic Stresses—2nd Edition” of Plants. Thank you very much for your time and consideration.
Reviewer 3
Comments and Suggestions for Authors
This paper entitled “ZnONPs Alleviates Salt Stress in Maize Seedlings by Improving Antioxidant Defense and Photosynthesis Potential” This study aimed to investigate the effects of ZnONPs on maize under salt stress. To explore this, two maize genotypes, NKY298-1 and NKY211, were analyzed for changes in phenotype, reactive oxygen species balance, membrane system condition, antioxidant activity, and photosynthetic capacity with different ZnONP applications under 120 mmol/L NaCl stress. The results showed that 120 mmol/L NaCl stress induced oxidative stress in both maize seedlings, with malondialdehyde (MDA), hydrogen peroxide (H2O2), and superoxide anion (O2•−) levels increasing by 24.81%, 67.27%, and 59.32%, respectively. This stress also affected membrane stability, reducing the membrane stability index (MSI) by 28.38%, and lowered the net photosynthetic rate (Pn) by 25% compared with the CK treatment. Importantly, the alleviating effects of various ZnONP concentrations differed among maize genotypes under NaCl stress.
This objective is highly relevant, given the increasing importance of developing salt-tolerant crops in the context of global soil salinization.
The authors used appropriate methodology and experimental design to test the study hypothesis, also.
I consider this topic to be relevant to the plants research field. It is an important work that could be helpful to researchers and appealing to readers of Plants.
The paper is well-organized, easily readable, and presented in a well-structured manner. The figures and tables are appropriate and easy to understand.
Thank you very much for your positive comments. We then have re-submitted the manuscript.
Thank you for your consideration.
- The bibliography used is well-documented, and the references are appropriate for the paper!
Thanks for your comments. We can guarantee that all the references cited in the manuscript are traceable. We then have re-submitted the manuscript.
Thank you for your consideration.
- It is not clear how ZnONPs are applied (in nutrient solutions? foliar treatment? exact concentrations tested). The number of concentrations tested was not specified, nor was it clear why two different genotypes were used.
Thanks for your comments. ZnONPs is dissolved in water and sprayed onto the leaf surface of maize seedlings. The concentration of NaCl is 120 mM, and there are five concentrations of ZnONPs, namely 10, 50, 100, 150, and 180 μmol/L. The two maize genotypes selected in the experiment are two salt-sensitive materials screened out by the team in previous experiments. So as to better observe the ability of ZnONPs to alleviate salt stress. We then have re-submitted the manuscript.
Thank you for your consideration.
- Please explain why you used a concentration of 120mM NaCl and why you used two genotypes for corn. Justify your answer. Please state the study hypothesis explicitly.
Thanks for your comments. The concentration of NaCl was the one that caused more harm to maize seedlings but was not lethal, which was screened out in previous experiments. The two maize genotypes selected in the experiment were two salt-sensitive materials screened out by the team in previous experiments. The combination of the two was to better observe the ability of ZnONPs to alleviate salt stress. We then have re-submitted the manuscript.
Thank you for your consideration.
- You state that "maize is affected by salt levels of more than 250 mM NaCl" but you are experimenting with 120 mM NaCl. The choice of this concentration needs to be explained (e.g., sublethal but strong enough to induce stress).
Thanks for your comments. The NaCl concentration selected in the experiment could cause sublethal maize seedlings but was sufficient to cause stress. However, a 250mM salt solution would directly cause the death of the seedlings. Based on the team's previous research, a 120mM salt solution was chosen as the experimental concentration. We then have re-submitted the manuscript.
Thank you for your consideration.
- "Causes a 65% yield loss" – this statement is very general; the context should be specified (at what salinity level? Under what conditions?). It is mentioned that “ZnONPs have been used successfully on other species (cucumber, rapeseed, sorghum, lupin), but not on corn”. However, the authors should specify why corn is a relevant model (e.g., strategic crop, sensitive to salinity, lack of studies with ZnONPs).
Thanks for your comments. As suggestion, we have supplemented the relevant content in lines 54-55. Besides, as your suggestion, we have added the relevant content about “why corn is a relevant model” in lines 85-87. We then have re-submitted the manuscript.
Thank you for your consideration.
- For the most part, the discussion reiterates what you have already stated in the Results section, with minor comparisons to the literature. A mechanistic interpretation is missing (why do ZnONPs decrease ROS? Is it due to catalytic action, role in Zn²⁺ homeostasis, or activation of zinc-dependent antioxidant enzymes?). The possibility that the effects of ZnONPs depend on the size, shape, and loading of the nanoparticles has not been discussed. It is stated that "ZnONPs alleviated oxidative stress" without discussing whether the effect is direct (interaction with ROS) or indirect (through increased antioxidant enzyme activity, osmotic or hormonal regulation).
Thanks for your comments. We fully agree with your suggestion, the ZnONPs mainly reduce oxidative stress by enhancing the activity of antioxidant enzymes, and its impact mainly depends on the concentration of ZnONPs rather than the size, shape and loading of nanoparticles (Tymoszuk et al. Zinc Oxide and Zinc Oxide Nanoparticles Impact on In Vitro Germination and Seedling Growth in Allium cepa L. Materials, 2020, 13). We then have re-submitted the manuscript.
Thank you for your consideration.
- Please specify in the text from the first appearance, what each abbreviated notation represents.
Thanks for your comments. As suggestion, we have checked the manuscript thought, and the relevant content has been improved. We then have re-submitted the manuscript.
Thank you for your consideration.
- The manuscript contains long and redundant sentences, and the phrasing is difficult to follow!
Thanks for your comments. As suggestion, we have checked and revised the language part of the entire manuscript to improve the overall quality of the manuscript as much as possible. We are very sorry for the inconvenience caused to you. We then have re-submitted the manuscript.
Thank you for your consideration.
- The authors should emphasize the novelty of their work
Thanks for your positive comments. As suggestion, we have supplemented the relevant content in the manuscript. We then have re-submitted the manuscript.
Thank you for your consideration.
- The limitations of this study, challenges and future perspectives should be included in the manuscript.
Thanks for your comments. As suggestion, we have supplemented the relevant content in the manuscript. We then have re-submitted the manuscript.
Thank you for your consideration.
- The authors should add a graphical abstract that highlights their work to a broader audience.
Thanks for your valuable suggestion, we fully agree with you. We understand that graphic abstract can help readers quickly grasp the core content of the paper. However, due to the current research conditions and time constraints, we are temporarily unable to provide high-quality graphic abstract. We will refine this part in subsequent research and supplement it when the manuscript is finally revised or submitted in the future. We then have re-submitted the manuscript.
Thank you for your consideration.
- Comments on the Quality of English Language Long sentences, cumbersome structure, repetitions. The sentences are sometimes awkward and redundant.
Thanks for your comments. As suggested, we have checked and revised the language part of the entire manuscript to improve the overall quality of the manuscript as much as possible. We are very sorry for the inconvenience caused to you. We then have re-submitted the manuscript.
Thank you for your consideration.
Best wishes!
Xiaoqiang Zhao Professor
State Key Laboratory of Aridland Crop Science, Gansu Agricultural University
- mail: zhaoxiaoq@gsau.edu.cn

Round 2
Reviewer 1 Report
Comments and Suggestions for Authors
Decision (Minor Revisions)
I have reviewed the revised manuscript and the authors’ responses to my earlier suggestions. The authors have made progress in clarifying the methodology, correcting figures and units, and refining the language. However, two critical issues remain inadequately addressed: (1) the lack of nanoparticle characterization (DLS, zeta potential, TEM, stability) and (2) the absence of ionic zinc (ZnSO₄) controls. These omissions limit the strength of the mechanistic interpretation.
If the journal is willing to accept the manuscript with these limitations, the Discussion should explicitly state that conclusions regarding nanoparticle-specific effects are tentative. Otherwise, additional experimental work would be necessary.
Overall, the revisions are a step forward, but they do not fully satisfy the most important points.
Minor Issues
- Statistical Reporting
- The Methods section mentions revisions, but it remains unclear which post-hoc test was applied after ANOVA.
- Corrections for multiple testing across 26 traits are not described, which could inflate Type I error.
- Sample Size Clarity
- Figures do not consistently report n (the number of plants per treatment, replicates per measurement). This information should be added to all figure captions.
- Abbreviations
- Terms such as MSI, CIE, and CAC appear abruptly. These should be defined upon first mention in both the Abstract and Methods.
- Terminology & Formatting
- Zea mays should always be italicized.
- Enzyme names should adhere to IUBMB conventions (e.g., “SOD” should be presented as “superoxide dismutase (SOD)”).
- Figures
- Scale bars are missing from all image/microscopy panels.
- A simple treatment timeline schematic would enhance clarity regarding the experimental design.
- Units and Consistency
- Double-check gas exchange units (they should be μmol CO₂ m⁻² s⁻¹).
- Some inconsistencies remain in the reporting of enzyme activity units.
- Writing Style
- Although improvements have been made, parts of the Abstract and Discussion are still too dense. Breaking them into shorter sentences would enhance readability.
- Discussion Depth
- The mechanistic interpretation is limited. More explanation of how ZnONPs may affect ROS scavenging, ion homeostasis, or enzyme activity would strengthen the narrative.
- Limitations (lack of nanoparticle characterization and absence of ionic control) should be clearly acknowledged.
- References
- Formatting inconsistencies remain; ensure all references are standardized to journal style and complete (including page numbers, DOI, etc.).
Abstract (lines 10–27)
- Line 10:
“Salt stress is one of the most serious environmental stresses that significantly inhibited maize growth and development, even seriously affected yield formation.”
→ “Salt stress is a significant environmental factor that inhibits maize growth and development, severely affecting yield formation.” - Lines 12–13:
“Interestingly, nanomaterials, especially ZnONPs can enhance various stresses resistance and maintain crops health growth.”
→ “Interestingly, nanomaterials, particularly ZnONPs, can enhance resistance to various stresses and support healthy crop growth.” - Line 14:
“However, the effects of ZnONPs on maize under salt stress remains unclear.”
→ “…the effects of ZnONPs on maize under salt stress remain unclear.” - Lines 21–23:
“…to rank the effectiveness of ZnONP doses. The findings suggest that 50–100 μM ZnONPs significantly mitigate salt damage, with optimal doses varying by genotype…”
→ “…to rank the effectiveness of ZnONP doses. The findings suggest that 50–100 μM ZnONPs significantly mitigate salt damage, with optimal doses differing by genotype…” - Line 24:
“Notably, the study's originality lies in its side-by-side genotype comparison and composite scoring across 26 traits.”
→ “…lies in its side-by-side genotype comparison and comprehensive composite scoring across 26 traits.”
Introduction (lines 31–73)
- Lines 32–33:
“It is estimated that approximately more than 800 million hectares of arable lands are affected by salinity (typically causing by the accumulation of NaCl) in the world, accounting for over 6% of the total lands.”
→ “It is estimated that over 800 million hectares of arable land worldwide are affected by salinity (typically caused by NaCl accumulation), accounting for over 6% of total land area.” - Line 34:
“…saline-alkali soils even account for 25% of arable lands in China and are underutilized.”
→ “…saline-alkali soils account for 25% of arable land in China and remain underutilized.” - Lines 38–39:
“…and 500 mM NaCl solution causes about 65% yield loss in wheat (Triticum aestivum L.).”
→ “…and a 500 mM NaCl solution causes approximately a 65% yield loss in wheat (Triticum aestivum L.).” - Line 41–42:
“Maize (Zea mays L.), a globally important cereal crop, it is generally used for food, feed, and bioenergy purposes, with an annual production of 1,200 million tons.”
→ “Maize (Zea mays L.) is a globally important cereal crop, widely used for food, feed, and bioenergy, with an annual production of 1,200 million tons.” - Lines 45–47:
“…meanwhile maize grown under NaCl stress disturbs ionic balance, which disorders plants’ mineral relation of Na+/K+ ratio and influence nutrients transport.”
→ “…meanwhile, maize grown under NaCl stress experiences disturbed ionic balance, disrupting the Na⁺/K⁺ ratio and influencing nutrient transport.” - Lines 66–68:
“…nano zinc oxide has higher specific surface activity, transparency or flexibility, and can also more precisely control ion release.”
→ “…nano-zinc oxide has higher surface activity, transparency, and flexibility, and allows for more precise control of ion release.”
These are only the first two sections. The full paper contains many similar small grammatical and stylistic issues (awkward phrasing, subject-verb agreement, missing articles, inconsistent use of “the,” long sentences that need breaking up).
Author Response
Dear Editor and Reviewers
Thank you for your letter of – and for the referee’s comments concerning our manuscript, “ZnONPs Alleviates Salt Stress in Maize Seedlings by Improving Antioxidant Defense and Photosynthesis Potential (Manuscript ID: plants-3867181)”. We have carefully studied these comments and have made corresponding corrections to the manuscript, which we describe in detail below. We would like to re-submit the manuscript and that for possible publication on the Special Issue: “Mitigation Strategies and Tolerance of Plants to Abiotic Stresses—2nd Edition” of Plants. Thank you very much for your time and consideration.
Reviewer 1
- I have reviewed the revised manuscript and the authors’ responses to my earlier suggestions. The authors have made progress in clarifying the methodology, correcting figures and units, and refining the language. However, two critical issues remain inadequately addressed: (1) the lack of nanoparticle characterization (DLS, zeta potential, TEM, stability) and (2) the absence of ionic zinc (ZnSO₄) controls. These omissions limit the strength of the mechanistic interpretation. If the journal is willing to accept the manuscript with these limitations, the Discussion should explicitly state that conclusions regarding nanoparticle-specific effects are tentative. Otherwise, additional experimental work would be necessary. Overall, the revisions are a step forward, but they do not fully satisfy the most important points.
Thanks for your comments. Thanks for your positive comments. To better clarify the properties of the nano zinc oxide, the scanning electron microscopy (SEM; S-3400N, Hitachi, Japan) analysis was performed, and the SEM image of the nano zinc oxide was shown in Fig. S1-1, which showed that the nano zinc oxide in our study had the average particle size 30±10 nm and displayed micropowders.
Fig. S1-1. The SEM image of nano zinc oxide in this study.
In this study, we conducted experiments by applying five concentrations of ZnONPs solutions to two maize genotypes seedlings (NKY298-1 and NKY211) at a concentration of 120 mM NaCl. It is expected that an appropriate concentration can be obtained to alleviate the damage caused by salt stress to maize. According to your meaningful suggestions, we will further explore the effects of conventional ZnSO4 and ZnONPs applications on salt stressed-maize in future experiments.
Thank you for your consideration.
- Statistical Reporting: The Methods section mentions revisions, but it remains unclear which post-hoc test was applied after ANOVA. Corrections for multiple testing across 26 traits are not described, which could inflate Type I error.
Thanks for your comments. As suggestion, we conducted a joint analysis of variance to verify 26 traits in the experiment, presented in table 1 form in 2.1 of the revised manuscript. Then we re-submitted the revised manuscript.
Thank you for your consideration.
- Sample Size Clarity: Figures do not consistently report n (the number of plants per treatment, replicates per measurement). This information should be added to all figure captions.
Thanks for your comments. The data shown in the figure are the average ± SD values of three sample replicates under different treatments.
Thank you for your consideration.
- Abbreviations: Terms such as MSI, CIE, and CAC appear abruptly. These should be defined upon first mention in both the Abstract and Methods.
Thanks for your comments. We once again checked the relevant content and ensured that abbreviations were accompanied by definitions when they appeared, such as MSI, CIE, and CAC in the abstract, etc. Then we submitted the manuscript.
Thank you for your consideration.
- Terminology & Formatting: Zea mays should always be italicized. Enzyme names should adhere to IUBMB conventions (e.g., “SOD” should be presented as “superoxide dismutase (SOD)”).
Thanks for your comments. As suggestion, we checked the relevant content once again.
Thank you for your consideration.
- Figures: Scale bars are missing from all image/microscopy panels. A simple treatment timeline schematic would enhance clarity regarding the experimental design.
Thanks for your comments. Most of the figures in the main text of the manuscript are bar charts without added scales. In the representation diagrams, scales of corresponding sizes have been added. At the same time, a processing schedule has been made and displayed in the Materials and methods section.
Thank you for your consideration.
- Units and Consistency: Double-check gas exchange units (they should be μmol CO₂ m⁻² s⁻¹). Some inconsistencies remain in the reporting of enzyme activity units.
Thanks for your comments. We checked the relevant content once again and made sure they were correct.
Thank you for your consideration.
- Writing Style: Although improvements have been made, parts of the Abstract and Discussion are still too dense. Breaking them into shorter sentences would enhance readability.
Thanks for your comments. Based on your suggestions, we have revised the relevant content in the manuscript.
Thank you for your consideration.
- Discussion Depth: The mechanistic interpretation is limited. More explanation of how ZnONPs may affect ROS scavenging, ion homeostasis, or enzyme activity would strengthen the narrative. Limitations (lack of nanoparticle characterization and absence of ionic control) should be clearly acknowledged.
Thanks for your positive comments. We revised the relevant content in the manuscript, and to better clarify the properties of the nano zinc oxide, the scanning electron microscopy (SEM; S-3400N, Hitachi, Japan) analysis was performed, and the SEM image of the nano zinc oxide was shown in Fig. S1-1, which showed that the nano zinc oxide in our study had the average particle size 30±10 nm and displayed micropowders.
Fig. S1-1. The SEM image of nano zinc oxide in this study.
In this study, we conducted experiments by applying five concentrations of ZnONPs solutions to two maize genotypes seedlings (NKY298-1 and NKY211) at a concentration of 120 mM NaCl. It is expected that an appropriate concentration can be obtained to alleviate the damage caused by salt stress to maize. According to your meaningful suggestions, we will further explore the effects of conventional ZnSO4 and ZnONPs applications on salt stressed-maize in future experiments. Then we re-submitted the revised manuscript.
Thank you for your consideration.
- References: Formatting inconsistencies remain; ensure all references are standardized to journal style and complete (including page numbers, DOI, etc.).
Thanks for your comments. We checked the format of the references and made sure it was correct. Then we re-submitted the manuscript.
Thank you for your consideration.
- Abstract (lines 10–27): Line 10:“Salt stress is one of the most serious environmental stresses that significantly inhibited maize growth and development, even seriously affected yield formation.”→ “Salt stress is a significant environmental factor that inhibits maize growth and development, severely affecting yield formation.”Lines 12–13: “Interestingly, nanomaterials, especially ZnONPs can enhance various stresses resistance and maintain crops health growth.”→ “Interestingly, nanomaterials, particularly ZnONPs, can enhance resistance to various stresses and support healthy crop growth.”Line 14: “However, the effects of ZnONPs on maize under salt stress remains unclear.”→ “…the effects of ZnONPs on maize under salt stress remain unclear.”Lines 21–23:“…to rank the effectiveness of ZnONP doses. The findings suggest that 50–100 μM ZnONPs significantly mitigate salt damage, with optimal doses varying by genotype…”→ “…to rank the effectiveness of ZnONP doses. The findings suggest that 50–100 μM ZnONPs significantly mitigate salt damage, with optimal doses differing by genotype…”Line 24:“Notably, the study's originality lies in its side-by-side genotype comparison and composite scoring across 26 traits.”→ “…lies in its side-by-side genotype comparison and comprehensive composite scoring across 26 traits.”
Introduction (lines 31–73): Lines 32–33:“It is estimated that approximately more than 800 million hectares of arable lands are affected by salinity (typically causing by the accumulation of NaCl) in the world, accounting for over 6% of the total lands.”→ “It is estimated that over 800 million hectares of arable land worldwide are affected by salinity (typically caused by NaCl accumulation), accounting for over 6% of total land area.”Line 34: “…saline-alkali soils even account for 25% of arable lands in China and are underutilized.” → “…saline-alkali soils account for 25% of arable land in China and remain underutilized.” Lines 38–39: “…and 500 mM NaCl solution causes about 65% yield loss in wheat (Triticum aestivum L.).” → “…and a 500 mM NaCl solution causes approximately a 65% yield loss in wheat (Triticum aestivum L.).” Line 41–42: “Maize (Zea mays L.), a globally important cereal crop, it is generally used for food, feed, and bioenergy purposes, with an annual production of 1,200 million tons.” → “Maize (Zea mays L.) is a globally important cereal crop, widely used for food, feed, and bioenergy, with an annual production of 1,200 million tons.” Lines 45–47: “…meanwhile maize grown under NaCl stress disturbs ionic balance, which disorders plants’ mineral relation of Na+/K+ ratio and influence nutrients transport.” → “…meanwhile, maize grown under NaCl stress experiences disturbed ionic balance, disrupting the Na⁺/K⁺ ratio and influencing nutrient transport.” Lines 66–68:
“…nano zinc oxide has higher specific surface activity, transparency or flexibility, and can also more precisely control ion release.” → “…nano-zinc oxide has higher surface activity, transparency, and flexibility, and allows for more precise control of ion release.” These are only the first two sections. The full paper contains many similar small grammatical and stylistic issues (awkward phrasing, subject-verb agreement, missing articles, inconsistent use of “the,” long sentences that need breaking up).
Thanks for your comments. We have revised the content of the first two parts according to your suggestions and also checked and modified the language of the entire manuscript. We are very sorry for the inconvenience caused to you. Then we re-submitted the revised manuscript.
Thank you for your consideration.
Best wishes!
Xiaoqiang Zhao Professor
State Key Laboratory of Aridland Crop Science, Gansu Agricultural University
- mail: zhaoxiaoq@gsau.edu.cn

Reviewer 2 Report
Comments and Suggestions for Authors
The manuscript has been improved, however there is still a lot of problems.
I cannot find any data, allowing authors to state in the abstract that “Notably, the study's originality 24 lies in its side-by-side genotype comparison and composite scoring across 26 traits”. There are no genotyping data - whatever is presented, is simply a phenotype.
There is still no information in the introduction, what does it mean “nanoparticle”. Authors added “Compared with micron and larger particle 66 zinc oxide, nano zinc oxide has higher specific surface activity, transparency or flexibility, 67 and can also more precisely control ion release” but the reader still does not know, what they mean by nano ZnO.
Even with new added information (catalogue number Yuanye Biotechnology Co., Ltd., Shanghai, China; CAS: 1314-13-2 402 V33683-100g ≥99%, powder, ≤100nm)) I still cannot find characteristics of the powder to be sure, that this is nano powder (e.g. how it was produced, what is surface characteristics). In the company webpage, there is no product under this catalogue number. Please provide actual physicochemical characterization, to assure that you used nano ZnO.
To claim, that the observed effect is the effect of nanopowder, there should be comparision with znO, that is not in the nanopowder formulation. Otherwise, it can be the effect of increased Zn concentration.
Significant part of the discussion is summary of the results, without actual discussion.
Comments on the Quality of English LanguageThe language is still a problem. Below, I'm providing some examples (not all) of what I mean.
line 20 „Then we propose”?
line 41, 43 - sentence construction
line 45 - style
line 46 - logic of sentence - maize grown under NaCL stress DOES NOT disturbee ionic balance. It should be e.g. “in maize grown under NaCl stress, there is disturbed ionic balance” - it really matters in English.
line 54 - logic of sentence
line 71, 403 - nanoparticle forms suspensions, not solutions. If there is a solution, there are no more nanoparticles.
line 145: How The contents of H2O2 and MDA can have catalogue numbers?
line 154 - logic of sentence (what is “various photosynthesis”?)
lines 158 to 163 - it is still not clear, what values in percent refer to. Content of chlorophyll is not measured in %.
line 164 to 166 - a decrease and increase in this sentence does not make sense
line 220 - the information of miR528 seems to not be related to the manuscript
line 259 - grammar or logic of the sentence, I’m not sure - how “two cluster method can be divided”? The method can lead to division, etc.
line 285 - “should be rather “we defined four parameters”. Definitions in this sentence makes no sense.
line 348 - I guess, “As” at the beginning is not necessary (or it was supposed to be “It is well known, that”. Otherwise, the sentence meaning lead to the conclusion, that the nanoparticles can do something, because they are well known.
line 406 - why the sentence starts with “soak”? There should be rather “The seed were soaked”.
line 416 - as line 406.
line 429 - why sentence strats with “and”? similar problem - line 491
Author Response
Reviewer 2
- I cannot find any data, allowing authors to state in the abstract that “Notably, the study's originality 24 lies in its side-by-side genotype comparison and composite scoring across 26 traits”. There are no genotyping data - whatever is presented, is simply a phenotype.
Thanks for your comments. In this study, two maize genotypes seedlings, including NKY298-1 and NKY211 were treated with seven treatments for seven days to measure the changes of 26 traits, i.e., seedling length (SL), root length (RL), seedling fresh weigh (SFW), root fresh weigh (RFW), O2•− content, H2O2 content, MDA content, NADP-ME content, PEPCK content, PPDK content, SOD activity, POD activity, CAT activity, APX activity, net photosynthetic rate (Pn), intercellular CO2 concentration (Ci), stomatal conductance (Gs), transpiration rate (Tr), chlorophyll a content (Ca), chlorophyll b content (Cb), carotenoids content (Car), relative content of chlorophyll (SPAD), chlorophyll a+b content : carotenoid content ratio (Cab/Car), chlorophyll a:b ratio (Ca/Cb), membrane stability index (MSI), and relative moisture content (RWC). As suggested, we have thus restated the corresponding contents were that “Notably, the study's originality lies in its side-by-side composite scoring across 26 traits across two maize genotypes seedlings” in the revised manuscript. We then have re-submitted the revised manuscript.
Thank you for your consideration.
- There is still no information in the introduction, what does it mean “nanoparticle”. Authors added “Compared with micron and larger particle 66 zinc oxide, nano zinc oxide has higher specific surface activity, transparency or flexibility, 67 and can also more precisely control ion release” but the reader still does not know, what they mean by nano ZnO.
Thanks for your comments. Nanoparticle, ultrafine unit with dimensions measured in nanometers (nm; 1 nm = 10−9 meter). Nanoparticles exist in the natural world and are also created as a result of human activities. Because of their submicroscopic size, they have unique material characteristics, and manufactured nanoparticles may find practical applications in a variety of areas, including medicine, engineering, catalysis, agricultural production, and environmental remediation (https://www.britannica.com/science/nanoparticle). Meanwhile, multiple previous studies have reported that nanoparticle can promote plant growth, and enhance stress resistance (Rajnandini, V.; Ajey, S.; Shubhra, K.; Pradeep, K. Alleviation of environmental stresses in crop plants by nanoparticles: recent advances and future. Journal of Biochemistry and Biotechnology, 2025, 34(3), 615-638. Singh, A.; Agrawal, S.; Rajput, V.D.; Ghazaryan, K.; Yesayan, A.; Minkina, T.; Zhao, Y.F.; Petropoulos, D.; Kriemadis, A; Papadakis, M.; Alexiou, A. Nnoparticles in revolutionizing crop production and agriculture to address salinity stress challenges for a sustainable future. Discover Applied Sciences, 2024, 6(6), 107831. W, H.H. Nanoparticles and plant adaptations to abiotic stresses. Functional Plant Biology, 2023, 50(11), 1-3. Pramanik, B.; Sar, P.; Bharti, R.; Gupta, R.; Purkayastha, S.; Sinha, S.; Chattaraj, S.; Mitra, D. Multifactorial role of nanoparticles in alleviating environmental stresses for sustainable crop production and protection. Plant Physiology and Biochemistry, 2023, 201, 107831.).
Though Zn nourishment of corn can improve consumer health, poor availability and low uptake (< 5% recovery) from conventional ZnSO4 fertilizers in alkaline soils (Ahmad et al., 2025.) (Ahmad, W.; Nepal, J.; Xin, X.P.; Nadeem, M.; He, Z.L. Nano zinc oxide enhances corn growth and nutrient uptake: comparison between soil drench and seed coating applications in alkaline sandy soils. Journal of Soil Science and Plant Nutrition, 2025, 25, 3198-3210.) necessitates exploring alternative Zn sources for crop biofortification. While zinc oxide nanoparticles (nano-ZnO), a typical nano-fertilizers, characterized by their small size, large surface area, and high charge density, demonstrate enhanced Zn uptake efficiency in crops (Asim et al., 2022; Thiruvengadam et al., 2018; Rizwan et al., 2017.) (Asim, M.; Ahmad, W.; Qamar, Z.; Awais, M.; Nepal, J.; Ahmad, I. Seed coating with zinc oxide nanofber (ZnONF) and urea improved zinc uptake; recovery efficiency, growth, and yield of bread wheat (Triticum aestivum L.). Journal of Soil Science and Plant Nutrition, 2022, 22(4), 5009-5020. Thiruvengadam, M.; Rajakumar, G.; Chung, I.M. Nanotechnology: current uses and future applications in the food industry. 3 Biotech, 2018, 8, 74. Rizwan, M.; Ali, S.; Qayyum, M.F.; Ok, Y.S.; Adrees, M.; Ibrahim. M.; Zia-ur-Rehman, M.; Farid, M.; Abbas, F. Effect of metal and metal oxide nanoparticles on growth and physiology of globally important food crops: A critical review. Journal of Hazard Materials, 2017, 322, 2-16.) and improved physiological and photosynthetic characteristics (Ahmad et al., 2023) (Ahmad, W.; Zou, Z.; Awais, M.; Munsif, F.; Khan, A.; Nepal, J.; Ahmad, M.; Akbar, S.; Ahmad, I.; Khan, M.S.; Qamar, Z.; Khan, H. Seed-primed and foliar oxozinc nanofiber application increased wheat production and Zn biofortification in calcareous-alkaline soil. Agronomy, 2023, 13, 400.).
As suggested, we have restated the corresponding contents were that “Compared with conventional ZnSO4 fertilizer, nano zinc oxide has distinctive physicochemical characteristics and increases the availability of micronutrients for plants.” We then have re-submitted the revised manuscript.
Thank you for your consideration.
- Even with new added information (catalogue number Yuanye Biotechnology Co., Ltd., Shanghai, China; CAS: 1314-13-2 402 V33683-100g ≥99%, powder, ≤100nm)) I still cannot find characteristics of the powder to be sure, that this is nano powder (e.g. how it was produced, what is surface characteristics). In the company webpage, there is no product under this catalogue number. Please provide actual physicochemical characterization, to assure that you used nano ZnO.
Thanks for your comments. To better clarify the properties of the nano zinc oxide, the scanning electron microscopy (SEM; S-3400N, Hitachi, Japan) analysis was performed, and the SEM image of the nano zinc oxide was shown in Fig. S1-1, which showed that the nano zinc oxide in our study had the average particle size 30±10 nm and displayed micropowders.
Fig. S1-1. The SEM image of nano zinc oxide in this study.
Thank you for your consideration.
- To claim, that the observed effect is the effect of nanopowder, there should be comparision with znO, that is not in the nanopowder formulation. Otherwise, it can be the effect of increased Zn concentration.
Thanks for your comments. Yes, I fully agree with your opinion on this issue. In this study, we conducted experiments by applying five concentrations of ZnONPs solutions to two maize genotypes seedlings (NKY298-1 and NKY211) at a concentration of 120 mM NaCl. It is expected that an appropriate concentration can be obtained to alleviate the damage caused by salt stress to maize. According to your meaningful suggestions, we will further explore the effects of conventional ZnSO4 and ZnONPs applications on salt stressed-maize in future experiments.
Thank you for your consideration.
- line 20 “Then we propose”?
Thanks for your comments. As suggestion, we have added “we” to the sentence. Then we submitted the relevant content.
Thank you for your consideration.
- line 41, 43 - sentence construction
Thanks for your comments. As suggestion, we have adjusted the sentence construction in the revised manuscript.
Thank you for your consideration.
- line 45 - style.
Thanks for your comments. We checked and adjusted the style of the relevant content.
Thank you for your consideration.
- line 46 - logic of sentence - maize grown under NaCL stress DOES NOT disturbee ionic balance. It should be e.g. “in maize grown under NaCl stress, there is disturbed ionic balance” - it really matters in English.
Thanks for your comments. As suggestion, we have adjusted the sentence structure in the revised manuscript.
Thank you for your consideration.
- line 54 - logic of sentence
Thanks for your comments. As suggestion, we have adjusted the logic of sentence. Then we re-submitted the revised manuscript.
Thank you for your consideration.
- line 71, 403 - nanoparticle forms suspensions, not solutions. If there is a solution, there are no more nanoparticles.
Thanks for your comments. We fully agree with your idea, then we revised the relevant content in the revised manuscript.
Thank you for your consideration.
- line 145: How The contents of H2O2 and MDA can have catalogue numbers?
Thanks for your comments. The contents of H2O2 and MDA were both determined using Solarbio's kits. The catalog number refers to the kits. Then we submitted the manuscript.
Thank you for your consideration.
- line 154 - logic of sentence (what is “various photosynthesis”?).
Thanks for your comments. As suggestion, we re-examined the literature and revised the relevant content. Then we re-submitted the revised manuscript.
Thank you for your consideration.
- lines 158 to 163 - it is still not clear, what values in percent refer to. Content of chlorophyll is not measured in %.
Thanks for your comments. The percentages in the manuscript refer to the degree of change in chlorophyll content under different treatments, rather than the content itself. Then we re-submitted the revised manuscript.
Thank you for your consideration.
- line 164 to 166 - a decrease and increase in this sentence does not make sense.
Thanks for your positive comments.
Thank you for your consideration.
- line 220 - the information of miR528 seems to not be related to the manuscript.
Thanks for your comments. As suggestion, we revised the relevant content in the manuscript.
Thank you for your consideration.
- line 259 - grammar or logic of the sentence, I’ m not sure - how “two cluster method can be divided”? The method can lead to division, etc.
Thanks for your comments. I'm very sorry that our statements might have caused you confusions. As suggested, we have restated the corresponding contents were that “Moreover, according to the results of both hierarchical clustering heatmap and circular clustering heat map, NKY298-1 (S), NKY211 (S), NKY211 (S+ ZnONPs-150), and NKY211 (S+ ZnONPs-180) were consistently divided into group IV, suggesting the growth status of the two maize genotypes was the worst under NaCl stress and the two concentrations of ZnONPs, resulting in the alleviation effect of the two concentrations on the damage caused by NKY211 under NaCl stress was very weak” in Lines 260-265 of the revised manuscript. We then have re-submitted the revised manuscript.
Thank you for your consideration.
- line 285 - “should be rather “we defined four parameters”. Definitions in this sentence makes no sense.
Thanks for your positive comments. As suggestion, we revised the relevant content.
Thank you for your consideration.
- line 348 - I guess, “As” at the beginning is not necessary (or it was supposed to be “It is well known, that”. Otherwise, the sentence meaning lead to the conclusion, that the nanoparticles can do something, because they are well known.
Thanks for your comments. As suggestion, we have revised the relevant content.
Thank you for your consideration.
- line 406 - why the sentence starts with “soak”? There should be rather “The seed were soaked”.
Thanks for your comments. As suggestion, we have revised the relevant content in the manuscript. Then we submitted the manuscript.
Thank you for your consideration.
- line 416 - as line 406.
Thanks for your comments. As suggestion, we have revised the relevant content in the manuscript. Then we submitted the manuscript.
Thank you for your consideration.
- line 429 - why sentence strats with “and”? similar problem - line 491.
Thanks for your comments. As suggestion, we have deleted the redundant text in relevant places.
Thank you for your consideration.
Best wishes!
Xiaoqiang Zhao Professor
State Key Laboratory of Aridland Crop Science, Gansu Agricultural University
- mail: zhaoxiaoq@gsau.edu.cn

Reviewer 3 Report
Comments and Suggestions for Authors
I'm satisfied by the changes made in the manuscript and Authors response and suggest to accept the manuscript in present form.
Author Response
Reviewer 3
I'm satisfied by the changes made in the manuscript and Authors response and suggest to accept the manuscript in present form.
Thanks for your positive comments!
Thank you for your consideration.
Best wishes!
Xiaoqiang Zhao Professor
State Key Laboratory of Aridland Crop Science, Gansu Agricultural University
- mail: zhaoxiaoq@gsau.edu.cn

Round 3
Reviewer 2 Report
Comments and Suggestions for Authors
The authors explained in their response what they understand as nanoZnO, but this information is still not in the introduction
The following should be added: “Nanoparticle, ultrafine unit with dimensions measured in nanometers (nm; 1 nm = 10−9 meter).”, with additional information about the surface stabilization of nanoZnO. This can be different - no coating, carboxyl acids, and other small chemicals. The actual size and coating change the final properties of the nanoparticle.
The quality of the TEM image provided in author's response does not allow to conclude that there are nanopowders. The scale bar is 10 μm; it is far too large to measure in nanometers.
Comments on the Quality of English LanguageThe manuscript really needs a review by a professional English proofreader - there are multiple small grammatical and logical mistakes that remain uncorrected.
Author Response
Dear Editor and Reviewers
Thank you for your letter of – and for the referee’s comments concerning our manuscript, “ZnONPs Alleviates Salt Stress in Maize Seedlings by Improving Antioxidant Defense and Photosynthesis Potential (Manuscript ID: plants-3867181)”. We have carefully studied these comments and have made corresponding corrections to the manuscript, which we describe in detail below. We would like to re-submit the manuscript and that for possible publication on the Special Issue: “Mitigation Strategies and Tolerance of Plants to Abiotic Stresses—2nd Edition” of Plants. Thank you very much for your time and consideration.
Reviewer 2
The authors explained in their response what they understand as nanoZnO, but this information is still not in the introduction
The following should be added: “Nanoparticle, ultrafine unit with dimensions measured in nanometers (nm; 1 nm = 10−9 meter).”, with additional information about the surface stabilization of nanoZnO. This can be different - no coating, carboxyl acids, and other small chemicals. The actual size and coating change the final properties of the nanoparticle.
The quality of the TEM image provided in author's response does not allow to conclude that there are nanopowders. The scale bar is 10 μm; it is far too large to measure in nanometers.
Thanks for your positive comments! We have added the relevant content to the corresponding position in the introduction. The specific products we use are shown in the following two pictures. (https://www.shyuanye.com/goods-V33683.html)
However, due to the precision limitations of the scanning electron microscopy (SEM; S-3400N, Hitachi, Japan) we used, only this size (10 μm) can be measured.
Thank you for your consideration.
Best wishes!
Xiaoqiang Zhao Professor
State Key Laboratory of Aridland Crop Science, Gansu Agricultural University
- mail: zhaoxiaoq@gsau.edu.cn
